# Learning-to-Optimize via Deep Unfolded Flows

**Augustinos D. Saravanos** [1] **Oswin So** [1] **H. M. Sabbir Ahmad** [2] **Chuchu Fan** [1]

## Abstract

We introduce **FlowOptimizer**, a deep unfolded, flow-based framework for learned iterative optimization. Motivated by the expressiveness of flow models, we represent each optimization iteration via a velocity field that operates on a population of candidate solutions, i.e., a set of parallel iterates, conditioned on contextual information including their objective values and gradients, as well as population-level statistics. The velocity field is initially trained in a simulation-free manner by matching displacements from source populations to improved target ones obtained through sampling the objective. Subsequently, we unfold this velocity field as the internal iteration of an optimization sequence, and fine-tune it in an end-to-end manner by directly optimizing objective values over a targeted class of problems. Notably, FlowOptimizer is a self-supervised framework whose training relies solely on objective evaluations without requiring knowledge of solutions. We evaluate our approach on a series of tasks from standard non-convex optimization benchmarks to real-world problems from supply chain, robotics and power grid applications. FlowOptimizer consistently outperforms well-established sampling-based/gradient-based traditional optimization and learning-to-optimize methods by orders of magnitude in terms of solution quality. We further highlight its ability to be trained on low-dimensional problems and successfully generalize to substantially higher-dimensional ($\times 10$) ones.

## 1. Introduction

Solving optimization problems efficiently lies at the core of modern engineering and machine learning applications.

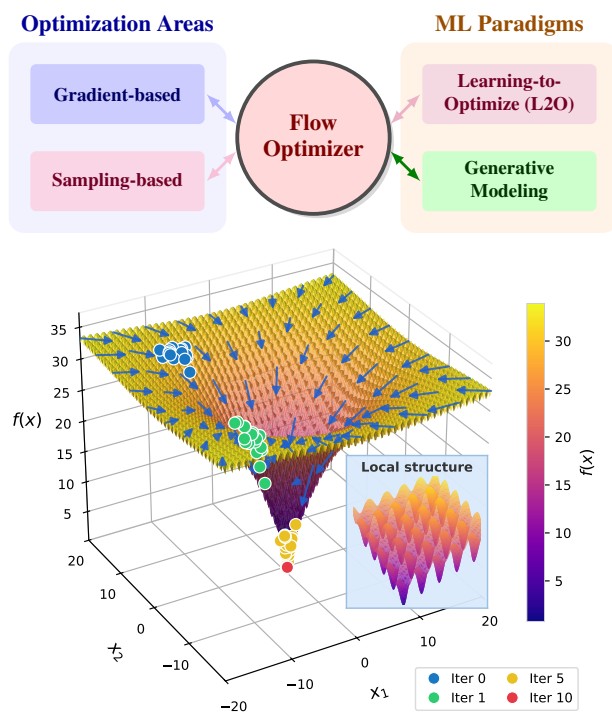

*Figure 1.* **Conceptual overview.** *Top:* From an optimization perspective, the proposed FlowOptimizer bridges gradient-based and sampling-based methods, while from a machine learning perspective, it lies at the intersection of learning-to-optimize and generative modeling. *Bottom:* FlowOptimizer represents each iteration via a learned velocity field conditioned on context information (objective values, gradients, statistics history, etc.), whose goal is to transport a population of candidate solutions to lower-cost regions. The shown example is the highly non-convex 2D Ackley function with an abundance of local minima. The arrows show the velocity field at iteration 5 w.r.t. to the current population/context.

Some of the most fundamental challenges arise from non-convexity leading to numerous local optima, as well as from the growing scale of the problems, which significantly increases their complexity and computational demands. At the same time, practical relevance demands methods that remain effective under limited computational budgets and generalize well across different problems without the need for extensive manual tuning.

Traditionally, optimization methodologies can be classified into two domains: gradient-based and sampling-based tech-

[1]Department of Aeronautics and Astronautics, Massachusetts Institute of Technology, Cambridge, MA, USA [2]Division of Systems Engineering, Boston University, Boston, MA, USA. Correspondence to: Augustinos D. Saravanos <asaravan@mit.edu>.

*Proceedings of the 43rd International Conference on Machine Learning*, Seoul, South Korea. PMLR 306, 2026. Copyright 2026 by the author(s).

Project website: https://mit-realm.github.io/flowopt/

niques. The former class exploits local derivative information to descend toward lower-cost regions (Wright et al., 1999), yet it is prone to getting trapped in poor local minima in non-convex settings. In contrast, sampling-based methods tend to be more robust to non-convexity (Kochenderfer & Wheeler, 2019), but typically face significant scalability challenges in high-dimensional regimes. Although there exists a notable amount of literature on hybrid approaches (Neri & Cotta, 2012; Wierstra et al., 2014), the pursuit of algorithms that effectively integrate gradient information and exploration via sampling remains an open challenge. Most importantly, striking an optimal balance between the two mechanisms is directly tied to the underlying problem geometry. Therefore, traditional optimization schemes with hand-crafted algorithmic choices naturally face difficulties to adapt across diverse optimization landscapes.

Learning-to-optimize (L2O) is an emerging paradigm which seeks to *learn optimization algorithm steps* directly from data (Chen et al., 2022; Shlezinger et al., 2022; Amos et al., 2023). By parameterizing the update rules of an optimizer and training them over a distribution of problems, L2O methods aim to capture problem structure and adapt algorithmic behavior beyond fixed, manually tuned schemes. Prior work has demonstrated substantially accelerated convergence and improved robustness across various types of optimization problems (Sambharya & Stellato, 2024; Saravanos et al., 2025; Oshin et al., 2026). However, much of the existing literature has primarily focused on learning purely gradient-based updates.

From a different angle, generative modeling has emerged the past years as a powerful framework for learning complex distributions, with flow-based models in particular gaining attention due to their stability, expressiveness and efficient training (Lipman et al., 2023; Liu et al., 2023). By learning continuous-time transformations between distributions, such models provide flexible and scalable mechanisms for modeling structured high-dimensional data.

Driven by the perspective that optimization can be viewed as seeking a distribution of points whose mass progressively concentrates in low-cost regions of the objective landscape, our key insight is that the expressiveness of flow-based models can be leveraged to *learn population-level optimization dynamics*. Rather than learning direct mappings to solutions or hand-designing update rules, we propose to model the optimization process itself as a learned transport of a population of candidate solutions, conditioned on their objective values, gradient information and population-level statistics. This viewpoint naturally enables a principled integration of gradient-driven exploitation and sampling-based exploration within a single, learnable framework that relies on velocity fields to guide the population towards lower-cost regions at each optimization iteration.

Specifically, our contributions are the following:

- We introduce **FlowOptimizer**, a novel learning-to-optimize framework whose core iteration is defined by a flow-based model that updates a population of candidate solutions conditioned on objective values, gradient information and statistics history. The proposed model can be pre-trained in a simulation-free manner using a flow matching objective, without requiring access to ground-truth optimal solutions.

- We further develop a deep unfolded formulation of FlowOptimizer by unrolling its iterative updates as layers of a deep neural network. This formulation enables end-to-end, self-supervised fine-tuning by directly minimizing a training loss defined in terms of objective evaluations, allowing the framework to adapt its dynamics across optimization iterations.

- Through extensive experimental evaluation, we demonstrate that FlowOptimizer consistently outperforms well-established gradient-based, sampling-based, and learning-to-optimize baselines across a range of highly non-convex optimization benchmarks, as well as real-world applications in robotics, power grid optimization, and supply chain problems. We further show that the dimension-agnostic variation of the proposed approach exhibits strong generalization capabilities to significantly higher-dimensional ($\times 10$) settings.

## 2. Related Work

An overview of related work in learning-to-optimize and generative modeling is provided below. For an extended discussion, we refer the reader to Appendix A.

**Learning-to-Optimize and Deep Unfolding.** Learning-to-optimize methodologies aim to learn optimization updates by training on problem data (Chen et al., 2022; Shlezinger et al., 2022; Amos et al., 2023). These approaches range from learning direct mappings from problem instances to solutions (Amos et al., 2023; Van Hentenryck, 2025) to more structured methods such as *deep unfolding*, which unrolls iterative optimization algorithms as sequential layers of a neural network, enabling end-to-end learning of algorithmic parameters (Monga et al., 2021; Zhang et al., 2020). This paradigm has led to significant advances in domains where classical optimization struggles with model mismatch or non-convexity. Recent works have further advanced deep unfolding by incorporating more expressive architectures and learning strategies (Sambharya & Stellato, 2024; Saravanos et al., 2025; Oshin et al., 2026), demonstrating improved performance and generalization across diverse optimization tasks.

**Generative Modeling for Optimization.** Generative models, such as diffusion models (Song et al., 2021) and normalizing flows (Lipman et al., 2023; Liu et al., 2023), have recently been explored for optimization by learning to generate solutions that minimize a given objective. While these models excel at capturing complex distributions, their direct application to end-to-end optimization remains challenging due to the implicit nature of the target distribution and the lack of ground-truth solutions in many settings. Nevertheless, several works have proposed leveraging flow-based or diffusion-based models for optimization, including model predictive control via flow matching (Kurtz & Burdick, 2025), solving KL-regularized optimization with diffusion models (Domingo-Enrich et al., 2025; So et al., 2026), and multi-robot trajectory optimization via flow matching (Idoko & Singh, 2025). Other recent efforts have sought to bridge generative modeling and optimization (Uehara et al., 2024; Kong et al., 2025; Mazé & Ahmed, 2023; Xie & Cheng, 2026), but the field remains nascent, with most approaches facing scalability and expressiveness limitations.

**Learning-to-Sample.** The problem of drawing samples from an unnormalized target distribution is closely related to optimization (Cheng, 2020). For example, by interpreting the unnormalized density as a Gibbs distribution associated with an objective function and a temperature parameter, samples increasingly concentrate around the global minimizers of the objective as the temperature decreases (Varadhan, 1984), a principle exploited by simulated annealing (Geman & Geman, 1984). Traditional sampling methods such as Markov Chain Monte Carlo (MCMC) (Geyer, 1992) are widely used, but often suffer from slow mixing in high dimensions or multi-modal landscapes (Łatuszyński et al., 2025). To address these challenges, recent works have explored learning neural components of samplers to accelerate convergence and improve sample quality (Zhang & Chen, 2022; Richter & Berner, 2024). However, many of these methods do not leverage information about the objective at multiple points simultaneously. Particle-based variational inference (Liu & Wang, 2016) offers an alternative by maintaining a population of interacting particles, with neural extensions proposed in (di Langosco et al., 2021).

# 3. FlowOptimizer: Leveraging a Flow-Based Model as the Core Optimization Mechanism

## 3.1. Problem Statement

We consider optimization problems of the form

$$\min_{x \in \mathbb{R}^n} f(x) \tag{1}$$

where $x \in \mathbb{R}^n$ is the decision variable and $f : \mathbb{R}^n \to \mathbb{R}$ is a potentially non-convex objective function to be minimized. We assume access to evaluations of $f(x)$ at arbi-

trary points $x$, and, when available, to gradient information $\nabla f(x)$. Given a class of objective functions $\mathcal{F}$, our goal is to learn an optimizer that can efficiently solve previously unseen problem instances with objectives $f \in \mathcal{F}$.

## 3.2. Flow Model as the Core Optimization Iteration

From a sampling point of view, optimization can be interpreted as seeking to sample from low-value regions of the cost landscape. Through this lens, we can view optimizing a class of objective functions as a generative modeling task that seeks a transport map from a simple initial distribution $\pi^0$ over candidate points to a complex target distribution $\pi^*$ whose mass concentrates on low-value regions of the objectives $f \in \mathcal{F}$. Nevertheless, directly employing diffusion models or flow-based models to learn solutions in an end-to-end manner is restrictive, as the underlying optimization dynamics are often too complex to be handled directly. Another major challenge is that the target distribution is only implicitly defined by the objectives $f \in \mathcal{F}$, and ground-truth solutions $x^* = \arg\min_x f(x)$ are likely to be unavailable in non-convex and/or large-scale settings, which fundamentally limits the applicability of supervised learning approaches from the generative modeling domain.

In the meantime, most algorithms within sampling-based optimization rely on iteratively updating a parametric distribution based on information coming from the current points population. Concretely, at iteration $k$, a population $\{x_j^k\}_{j=1}^P$, where $P$ is the population size, is sampled from a parametric distribution $q_{\theta^k}$, whose parameters are iteratively updated based on information extracted from the current population. Nevertheless, such algorithms often prove to be insufficient because these parametric distributions might lack the expressiveness as well as the contextual inputs required to perform optimization effectively, especially at a population level. Motivated by these complementary perspectives, our aim is to *learn* a flow-based model that acts as an *iterative optimizer* operating on a *population* of candidate solutions.

**Population-Level Update via Flows.** Specifically, we wish to learn a model $\mathcal{T}(\cdot)$ that transforms populations of points $\mathcal{X}^k = \{x_j^k\}_{j=1}^P$ to new populations $\mathcal{X}^{k+1} = \{x_j^{k+1}\}_{j=1}^P$ that correspond to lower cost objectives, i.e.,

$$\mathcal{X}^{k+1} = \mathcal{T}(\mathcal{X}^k, c^k) \tag{2}$$

where $c^k \in \mathcal{C}$ denotes contextual information available at iteration $k$. At its most basic form, the context variable $c^k$ includes the per-point objective values $f(\mathcal{X}^k) = \{f(x_j^k)\}_{j=1}^P$, gradients $\nabla f(\mathcal{X}^k) = \{\nabla f(x_j^k)\}_{j=1}^P$, as well as distributional statistics such as the empirical mean $\mu^k = \frac{1}{P}\sum_{j=1}^P x_j^k$ and covariance $\Sigma^k = \frac{1}{P}\sum_{j=1}^P (x_j^k - \mu^k)(x_j^k - \mu^k)^\top$ of the current population.

We parameterize the transformation $\mathcal{T}(\cdot)$ as the flow induced

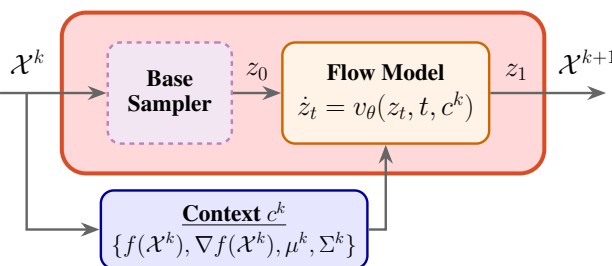

**FlowOptimizer Core Iteration** $k$

*Figure 2.* **The core optimization iteration of FlowOptimizer.** At iteration $k$, the current population $\mathcal{X}^k$ is optionally processed by a base sampler and then evolved through a flow model. The flow integrates population-level dynamics conditioned on contextual information $c^k$, producing the updated population $\mathcal{X}^{k+1}$.

by a continuous-time velocity field defined directly over the population. Specifically, we treat the population as a single state $z_t \in \mathbb{R}^{P \times n}$ evolving according to

$$\dot{z}_t = v_\theta(z_t, t, c^k), \qquad z_{t=0} = \mathcal{X}^k, \qquad (3)$$

where $v_\theta : \mathbb{R}^{P \times n} \times [0, 1] \times \mathcal{C} \to \mathbb{R}^{P \times n}$ is a learnable, context-conditioned velocity field parameterized by $\theta$. During inference, the population update is obtained by integrating and assigning $\mathcal{X}^{k+1} = z_{t=1}$. Importantly, the velocity field acts jointly on the entire population, thus benefiting from capturing population-level interactions.

**Permutation Invariance via Self-Attention.** Since the population $\mathcal{X}^k$ is an unordered set, the learned velocity field $v_\theta$ must be permutation equivariant with respect to the population members. Formally, for any permutation $\pi$ acting on the population indices, we require

$$v_\theta(\pi z_t, t, \pi c_{\text{loc}}^k, c_{\text{glob}}^k) = \pi\, v_\theta(z_t, t, c_{\text{loc}}^k, c_{\text{glob}}^k), \quad (4)$$

where the context $c^k$ is decomposed into the (local) member-level information part $c_{\text{loc}}^k = \{ c_j^k \}_{j=1}^P$, with $c_j^k = \{f(x_j^k), \nabla f(x_j^k)\}$, and the (global) population-level part $c_{\text{glob}}^k = \{\mu^k, \Sigma^k, \bar{g}^k\}$. To achieve permutation equivariance, we parameterize $v_\theta$ using a self-attention architecture that operates jointly on the population state and the local context. Letting $\phi_z(\cdot)$ and $\phi_c(\cdot)$ denote shared embedding maps, we form joint per-member representations

$$h_j = \big[\, \phi_z(z_t^{(j)}),\ \phi_c(c_j^k)\,\big], \qquad j = 1, \ldots, P, \quad (5)$$

and apply a permutation-equivariant self-attention operator

$$(\tilde{h}_1, \ldots, \tilde{h}_P) = \text{SelfAttention}(h_1, \ldots, h_P), \quad (6)$$

which enables information exchange across the population while preserving equivariance. The global features $c_{\text{glob}}^k$, along with the flow time $t$, are incorporated through shared

inputs across all population members. The resulting architecture yields a permutation equivariant velocity field, while remaining expressive enough to capture both inter-member interactions and population-level information. We further emphasize the flexibility of the model to use different population sizes during training and deployment.

**Base Sampler.** An optional base sampling operator $\mathcal{S}(\cdot)$ may be applied to the current population $\mathcal{X}^k$ to generate the initial flow state $z_{t=0}$, i.e., $z_{t=0} = \mathcal{S}(\mathcal{X}^k)$, instead of directly setting $z_{t=0} = \mathcal{X}^k$. When a base sampler is used, the original population $\mathcal{X}^k$ is appended to the context information $c^k$. A straightforward choice is a Gaussian sampler, where new candidates are drawn from a normal distribution fitted to the current population, in the spirit of classical sampling-based methods such as the Cross-Entropy Method (CEM) (De Boer et al., 2005). Crucially, in contrast to such methods, the Gaussian distribution is used solely as a source distribution, while the subsequent population transformation is driven by the expressive learned flow model. Empirically, we observe that in some cases incorporating a base sampler can stabilize the optimization process by inducing a more controlled exploration around the current population. However, incorporating such simple parametric samplers may also restrict population diversity in highly multimodal landscapes. By keeping the base sampler optional, our framework can leverage its stabilizing effect if needed, while relying on the flow model to capture richer population transformations.

### 3.3. Dimension-Agnostic Variation

To enable the same model to be applicable to different population and problem dimensions, we also present a dimension-agnostic parameterization of the velocity field. Given a population state $z_t \in \mathbb{R}^{P \times n}$ and gradients $\nabla f(\mathcal{X}^k) \in \mathbb{R}^{P \times n}$, we first construct fixed-size *per-member tokens* by pooling statistics across the coordinate dimension, e.g.,

$$s_j(t) = \big[\text{Stat}(z_{t,j}), f(x_j^k), \text{Stat}(\nabla f(x_j^k)), t\big] \in \mathbb{R}^m, \ (7)$$

which is mapped through a shared embedding network to obtain $h_j \in \mathbb{R}^H$. The statistics pooling operator is given by $\text{Stat}(\cdot) = [\text{Mean}(\cdot), \text{Std}(\cdot), \text{Max}(\cdot)]$. We then apply self-attention over $\{h_j\}_{j=1}^P$ to capture population-level interactions while remaining agnostic to $n$. To generate velocities, we use a *shared per-dimension decoder* applied independently across coordinates:

$$v_{j,i}^{\text{loc}} = \psi_\theta\big(z_{t,j,i},\ [\nabla f(x_j^k)]_i,\ h_j,\ t\big), \quad i = 1, \ldots, n, \ (8)$$

which enforces parameter sharing across dimensions and yields a coordinate-wise equivariant update. Note that due to the use of shared embeddings and permutation-invariant self-attention, the resulting representation is also independent of

**Algorithm 1** Pre-training phase via flow matching loss

1: **Input:** Model $v_\theta$, objective class $\mathcal{F}$, population size $P$, batch size $B$, improvement operator $\text{Improve}(\cdot, f)$
2: **for** $b = 0$ to $B - 1$ **do**
3:     Select objective function $f_b \in \mathcal{F}$
4:     Sample source population $\mathcal{X}_b = \{x_0^j\}_{j=1}^P \sim p_0$
5:     Compute context $c_b \leftarrow \text{Context}(\mathcal{X}_b, f_b(\mathcal{X}_b), \dots)$
6:     Construct target population $\mathcal{X}_b^+ \leftarrow \text{Improve}(\mathcal{X}_b, f_b)$
7:     Sample pairs $(z_0, z_1) \sim \mathcal{U}(\mathcal{X}_b \times \mathcal{X}_b^+)$
8:     Sample $t \sim \mathcal{U}[0, 1]$ and set $z_t \leftarrow (1 - t)z_0 + t z_1$
9:     Compute loss $\mathcal{L}_{\text{FM}}(\theta) \leftarrow \|v_\theta(z_t, t, c_b) - (z_1 - z_0)\|_2^2$
10:     Update parameters $\theta \leftarrow \theta - \eta \nabla_\theta \mathcal{L}_{\text{FM}}(\theta)$
11: **end for**

the population size $P$. Finally, to capture cross-dimensional coupling without introducing $n$-dependent parameters, we optionally add a low-rank mixing term

$$v_{j,i} = v_{j,i}^{\text{loc}} + \langle a_{j,i}, b_j \rangle, \qquad a_{j,i}, b_j \in \mathbb{R}^r,$$

where $r \ll n$ and both $a_{j,i}$ and $b_j$ are produced by shared networks from $(z_{t,j,i}, h_j, t)$ (and gradients when available). This design yields a velocity field $v_\theta(z_t, t, c^k) \in \mathbb{R}^{P \times n}$ whose parameter count does not scale with $n$, facilitating generalization across dimensions.

### 3.4. Connections to classical and learned optimizers

The representation employed by the FlowOptimizer is sufficiently expressive to recover several classical and learned optimization paradigms as special cases. When a Gaussian base sampler is used, the resulting population dynamics closely resemble sampling-based methods such as CEM and related evolutionary strategies. In contrast, when no base sampler is employed and the population size is reduced to one, the update rule recovers a deterministic, gradient-driven iteration, closely aligning with classical first-order methods as well as learning-to-optimize approaches that learn parametric update rules. When the population size is greater than one but no base sampler is used, the deterministic interactions among candidate solutions bear resemblance to particle swarm optimization (PSO), where particles update their states by aggregating information from their local history and the collective population. Beyond these limiting cases, FlowOptimizer enables rich interactions among multiple candidate solutions through learned population-level dynamics, allowing it to effectively blend the complementary strengths of sampling-based global exploration and gradient-based local refinement.

### 3.5. Pre-Training via Flow Matching

We initially train a single model instance (i.e., one iteration) $\mathcal{T}(\cdot)$ of the flow optimizer using the flow matching loss due to its training efficiency. In particular, we

train the velocity field $v_\theta$ to match displacements from randomly drawn source population points $\mathcal{X}$ to improved ones $\mathcal{X}^+ = \text{Improve}(\mathcal{X}, f)$ in terms of their objective values. The improved population is obtained by sampling $\gamma P$ candidates, with $\gamma \gg 1$, in a neighborhood of $\mathcal{X}$ and retaining the best $P$ points according to $f$. Given pairs $(z_0, z_1)$ with $z_0 \in \mathcal{X}$ and $z_1 \in \mathcal{X}^+$, we define an interpolation

$$z_t = (1 - t)z_0 + t z_1, \qquad t \sim \mathcal{U}[0, 1], \qquad (9)$$

and train the velocity field to predict the displacement $\Delta z = z_1 - z_0$ along this path, conditioned on the context information. The flow matching training loss is then

$$\mathcal{L}_{\text{FM}} = \mathbb{E}_{t, z_0, z_1} \|v_\theta(z_t, t, c) - \Delta z\|_2^2 \qquad (10)$$

where the expectation is taken over objectives $f \sim \mathcal{F}$, source populations $\mathcal{X} \sim p_0$, improvement operators $\text{Improve}(\cdot, f)$, uniformly sampled pairs $(z_0, z_1) \in \mathcal{X} \times \mathcal{X}^+$, and interpolation times $t \sim \mathcal{U}[0, 1]$. This pre-training phase yields a single learned update operator $\mathcal{T}(\cdot)$ that captures a generic improvement direction across a class of objectives, without requiring access to ground-truth solutions. The pre-training process is detailed in Alg. 1.

## 4. Deep Unfolded FlowOptimizer

The full proposed architecture emerges by unfolding the FlowOptimizer and training in an end-to-end manner. In Section 4.1, we introduce the deep unfolded architecture which unrolls the iterations of the FlowOptimizer as sequential layers. In Section 4.2, we then describe an end-to-end fine-tuning procedure in which all unfolded layers are jointly optimized by directly minimizing the objective values induced by the population iterates.

### 4.1. Unfolding the Flow Model as an Iterative Optimizer

We unfold the FlowOptimizer for a prescribed number of iterations $K$, each iteration as sequential layers in our architecture (Fig. 3). To accommodate the distinct roles played by different stages of the optimization process, we allow for each flow model to vary across iterations, resulting in iteration-specific models $\mathcal{T}^k$, $k = 0, \dots, K - 1$, with corresponding velocity fields $v_{\theta^k}(z_t^k, t, c^k)$. Each iteration is initialized with the same pre-trained model obtained via flow matching. Furthermore, to improve steady-state behavior, we share parameters across the final $K_{\text{final}}$ iterations by repeating the same velocity model across the iterations $K - K_{\text{final}}, \dots, K - 1$. This design both stabilizes the terminal phase of the optimization and enables the unfolded network to be executed for additional iterations at inference time, if desired, without introducing new parameters.

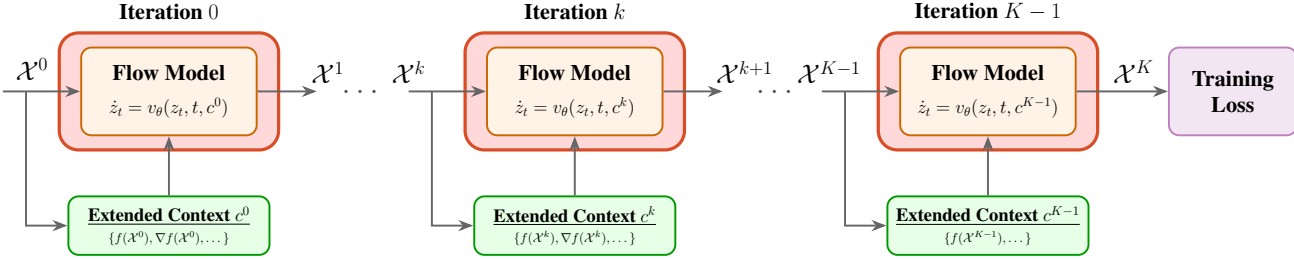

*Figure 3.* **The full deep unfolded FlowOptimizer architecture .** By unrolling the iterations of FlowOptimizer as sequential layers in a deep neural network, we obtain the final architecture which is trained in an end-to-end fashion via an unsupervised training-loss.

---

**Algorithm 2** Fine-tuning phase via deep unfolding

---

1: **Input:** Model $v_\theta$, objective class $\mathcal{F}$, population size $P$, unrolling depth $K$, batch size $B$
2: **for** $b = 0$ to $B - 1$ **do**
3:   Select objective function $f_b \in \mathcal{F}$
4:   Sample initial population $\mathcal{X}_0 = \{x_0^j\}_{j=1}^P \sim p_0$
5:   **for** $k = 0$ to $K - 1$ **do**
6:     Get context $c_k \leftarrow \text{Context}(\mathcal{X}_k, f_b(\mathcal{X}_k), \dots)$
7:     Set $z_0 \leftarrow \mathcal{X}_k$
8:     $z_1 \leftarrow \text{IntegrateODE}(\dot{z} = v_\theta(z_t, t, c_k), z_0)$
9:     Set $\mathcal{X}_{k+1} \leftarrow z_1$
10:    Compute current iteration loss $\ell_{k+1}(\theta) \leftarrow \alpha\text{Best}(f_b(\mathcal{X}_{k+1})) + (1 - \alpha)\text{Mean}(f_b(\mathcal{X}_{k+1}))$
11:   **end for**
12:   Compute total loss $\mathcal{L}_{\text{FT}}(\theta) \leftarrow \sum_{k=1}^K w_k \ell_k(\theta)$
13:   Update parameters $\theta \leftarrow \theta - \eta \nabla_\theta \mathcal{L}_{\text{FT}}(\theta)$
14: **end for**

---

### 4.2. End-to-End Fine-Tuning

Subsequently, we fine-tune the deep unfolded model using a training loss that directly minimizes the objective values $f(\mathcal{X}^k)$. To obtain each new iterate $\mathcal{X}^{k+1}$, we perform Euler integration on the velocity field ODEs,

$$z_{t+\Delta t}^k = z_t^k + v_\theta(z_t, t, c^k)\Delta t \tag{11}$$

where $\Delta t$ denotes the discretization step.

Then for training, we define the per-iteration loss

$$\ell_k(\mathcal{X}^k) = \alpha\text{Best}(f(\mathcal{X}_k)) + (1 - \alpha)\text{Mean}(f(\mathcal{X}_k)) \tag{12}$$

where $\text{Best}(f(\mathcal{X}_k)) := \text{SoftMin}_{j=1,\dots,P} f(x_j^k)$ is an approximation of the best objective achieved by a point within $\mathcal{X}^k$, $\text{Mean}(f(\mathcal{X}_k)) := \frac{1}{P}\sum_{j=1}^P f(x_j^k)$ is the average objective value of the population $\mathcal{X}^k$, and $\alpha \in [0, 1]$ is a balancing parameter between these two terms. Then, the end-to-end training loss is given by

$$\mathcal{L}_{\text{FT}} = \sum_{k=1}^K w_k \ell_k(\mathcal{X}^k) \tag{13}$$

where the weights $w_k, k = 1, \dots, K$ assign each iteration $k$ a specific amount of importance. The fine-tuning process is detailed in Alg. 2.

## 5. Experiments

We demonstrate the performance of `FlowOptimizer` on a diverse set of problems, ranging from standard optimization benchmarks to real-world applications. Section 5.1 summarizes several sampling-based/gradient based optimization or L2O methodologies used as baselines. In Section 5.2 we compare the methods on various standard optimization benchmarks, while in Section 5.3, we extend this comparison on real-world supply chain, power grid and robotics problems. In Section 5.4, we demonstrate that the proposed dimension-agnostic variant can be trained on low-dimensional problems and then successfully deployed on substantially higher-dimensional ones. Finally, an overall discussion and potential limitations are provided in Section 5.5. For all methods, we use a population of size $P = 20$. The full experiment details are given in Appendix B.

### 5.1. Baselines

We compare `FlowOptimizer` against sampling-based methods including `Random Search (RS)`, `Cross Entropy Method (CEM)` (De Boer et al., 2005), `Covariance Matrix Adaptation Evolution Strategy (CMA-ES)` (Hansen & Ostermeier, 2001), and `Particle Swarm Optimization (PSO)` (Kennedy & Eberhart, 1995), gradient-based techniques with multi-start (MS) randomization (Zhigljavsky, 2025) including `MS Gradient Descent (MS-GD)`, `MS Nesterov Accelerated GD (MS-NAG)`, and `MS Adam (MS-Adam)`, as well as a multi-start variation of `Learning-to-Optimize Gradient Descent (L2O-GD)` (Sambharya & Stellato, 2024). A full description of all baselines is given in Appendix B.4.

### 5.2. Standard Optimization Benchmarks

We start with evaluating the performance of all methods on a series of popular non-convex optimization benchmarks. In

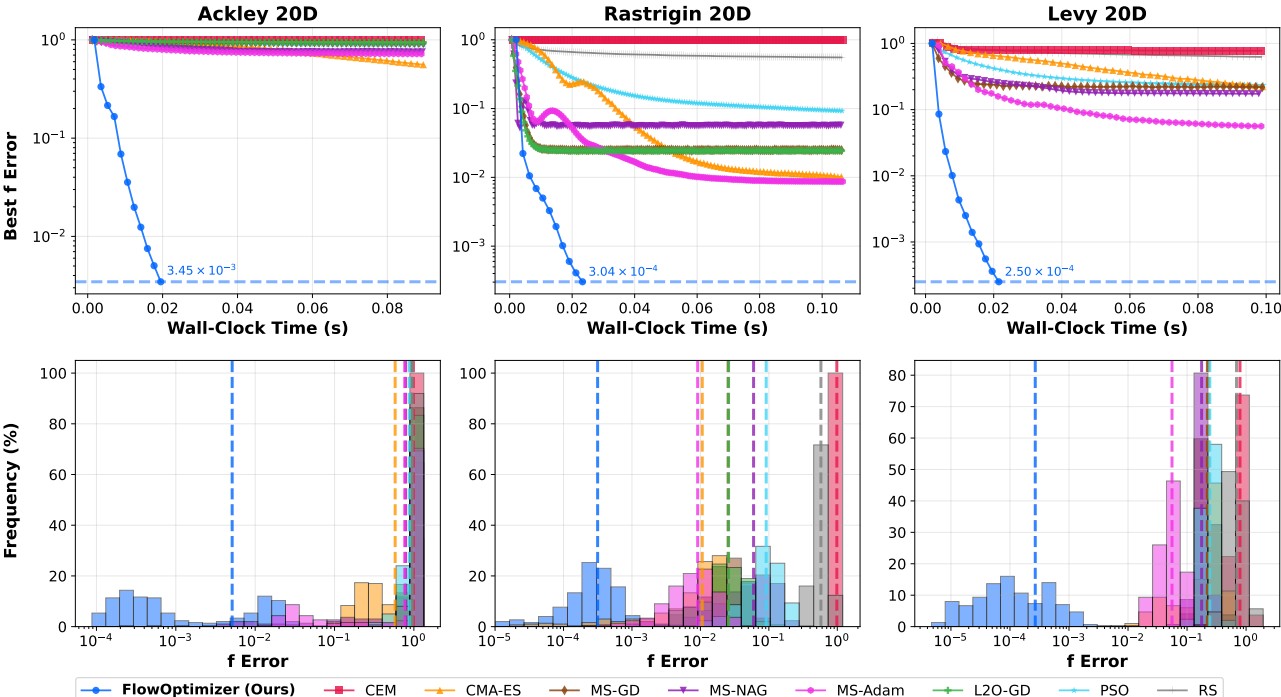

*Figure 4.* **Standard Optimization Benchmarks.** Comparison of metric $|\text{Best}(f(\mathcal{X}^k)) - f^*| / |\text{Best}(f(\mathcal{X}^0)) - f^*|$ between all methods on three standard optimization benchmarks (Ackley 20D, Rastrigin 20D and Levy 20D). The top row reports the average across all evaluation problem instances. The bottom row reports the distribution of final errors. The proposed FlowOptimizer consistently outperforms all baselines. The baseline methods are allowed to run for $\times 5$ more wall-clock time.

particular, the objectives we consider are the Ackley, Rastrigin, and Levy functions, all of which present highly non-convex and multimodal cost landscapes that are especially hard to be handled by either sampling-based or gradient-based methods. For all problems, we consider a dimension $n = 20$. The problems are fully detailed in Appendix B.5.

Figure 4 compares convergence speed and final solution quality across all methods on three standard benchmarks. The top row reports the average best objective error over wall-clock time, normalized by the initial error, while the bottom row shows the distribution of final errors across problem instances. **FlowOptimizer** consistently achieves the lowest objective error across all three benchmarks, converging rapidly to near-optimal solutions well within the allotted time budget, despite the baselines being allowed $\times 5$ more wall-clock time. On all three benchmarks, **FlowOptimizer** reaches final errors orders of magnitude below those of all baselines. The histogram panels reveal that **FlowOptimizer** not only attains lower errors on average, but does so with markedly tighter distributions, indicating greater reliability across random initializations and problem instances. In contrast, CEM and RS stagnate near the initial error on all benchmarks, reflecting their poor scalability to high-dimensional multimodal landscapes.

CMA-ES shows moderate progress on Rastrigin but fails to consistently improve on Ackley and Levy. Gradient-based multi-start methods (MS-GD, MS-NAG, MS-Adam) and PSO exhibit high variance and tend to concentrate probability mass at suboptimal error levels, suggesting susceptibility to local minima. L2O-GD, despite leveraging learned update rules, similarly fails to match **FlowOptimizer**, highlighting that learning individual GD steps is insufficient without explicit population-level coordination and a richer representation.

### 5.3. Real-World Optimization Problems

We then evaluate our method on real-world robotics, power grid and supply chain problems. Table 1 reports the average ratio $\text{Best}(f(\mathcal{X}^k))/f(\mu^0)$ across all methods, where lower values indicate better optimization performance relative to the initial solution. Several consistent trends emerge across problem classes and dimensions.

**Robotic Arm Inverse Kinematics.** For the inverse kinematics task, all methods benefit from the relatively smooth structure of the objective at low dimension ($n = 10$), with CMA-ES already achieving strong performance (0.008). However, as the dimension increases to $n = 30$, both sampling-based and gradient-based baselines exhibit no-

*Table 1.* **Real-World Optimization Problems.** Average Ratio Best$(f(\mathcal{X}^k))/f(\mu^0)$ across different methods and problem classes.

| Problem Class | Dim. $n$ | **FlowOpt (ours)** | RS | CEM | CMA-ES | MSGD | MSNAG | MS-Adam | L2O-GD | PSO |
|---|---|---|---|---|---|---|---|---|---|---|
| Robotic Arm | 10 | **0.003** | 0.18 | 0.09 | 0.008 | 0.014 | 0.021 | 0.009 | 0.012 | 0.005 |
| | 30 | **0.007** | 0.31 | 0.18 | 0.04 | 0.18 | 0.17 | 0.07 | 0.12 | 0.011 |
| Power Grid | 20 | **5e-4** | 0.53 | 0.14 | 0.03 | 0.05 | 0.03 | 0.007 | 0.03 | 0.02 |
| | 50 | **1.2e-3** | 0.83 | 0.31 | 0.11 | 0.23 | 0.17 | 0.05 | 0.08 | 0.09 |
| Supply Chain | 40 | **0.007** | 0.47 | 0.09 | 0.05 | 0.07 | 0.03 | 0.02 | 0.013 | 0.02 |
| | 80 | **0.013** | 0.68 | 0.14 | 0.06 | 0.09 | 0.03 | 0.02 | 0.027 | 0.03 |

ticeable degradation, with `CMA-ES` deteriorating to 0.04 and gradient-based methods remaining above 0.12. `PSO` emerges as the closest competitor across both settings (0.005 and 0.011). `FlowOptimizer` maintains a substantially lower error across both settings (0.003 and 0.007), indicating improved scalability and robustness to increasing redundancy and obstacle-induced non-convexity.

**Power Grid Control.** The Economic Load Dispatch problem (Sinha et al., 2003) presents a highly challenging landscape due to valve-point effects and pronounced multimodality. Here, purely gradient-based methods consistently converge to suboptimal solutions, while sampling-based approaches provide improved robustness but still suffer from scaling limitations as $n$ increases from 20 to 50. Among the baselines, `MS-Adam` achieves the strongest performance at $n = 20$ (0.007). Overall, `FlowOptimizer` achieves orders-of-magnitude improvements over all baselines, reducing the error ratio to $5 \times 10^{-4}$ and $1.2 \times 10^{-3}$, respectively. This gap highlights the benefit of integrating gradient information with population-level transport dynamics in highly rugged objective landscapes.

**Supply Chain Optimization.** In the Multi-Source Weber problem (Brimberg et al., 2008), the presence of combinatorial symmetries and clustering ambiguities leads to multiple near-optimal basins. At $n = 40$, `MS-Adam` and `MS-NAG` remain competitive among the baselines (0.02 and 0.03), while `CEM` and classical gradient methods perform worse. At $n = 80$, however, `CMA-ES` (0.06) provides the strongest baseline performance as gradient-based methods fail to scale. `FlowOptimizer` consistently achieves the lowest error across both $n = 40$ and $n = 80$ (0.007 and 0.013), suggesting that the learned flow dynamics search effectively in symmetric, non-convex landscapes.

**Overall Results.** Across all real-world benchmarks, `FlowOptimizer` consistently outperforms both sampling/gradient-based classical optimization methods and learning-based gradient baselines, with performance gaps that widen as problem dimension increases. These results indicate that learning population-level optimization dynamics enables improved scalability and

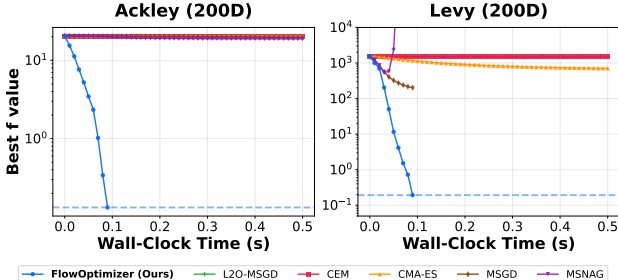

*Figure 5.* **Applying FlowOptimizer (dimension-agnostic variant) to higher-dimensional problems.** Models trained on 20D problems are successfully deployed on 200D problems, where the performance of competing baselines degrades significantly.

robustness compared to relying exclusively on traditional gradient-based or sampling-based paradigms.

### 5.4. Generalization to Higher-Dimensional Problems

A well-known limitation of learning-to-optimize methods is their difficulty to generalize beyond the problem dimensions seen during training. We evaluate this challenge using the dimension-agnostic variant of `FlowOptimizer` trained exclusively on 20-dimensional problems and deployed without retraining on 200-dimensional instances of highly non-convex benchmarks, including Ackley and Lévy functions. As shown in Fig. 5, `FlowOptimizer` retains good performance despite the tenfold increase in dimensionality. In contrast, gradient-based, sampling-based, and learning-to-optimize baselines exhibit severe performance degradation and often fail to make meaningful progress within the same wall-clock budget. These results highlight that the proposed flow-based population dynamics enable strong generalization without dimension-specific tuning.

### 5.5. Discussion

**What makes `FlowOptimizer` a stronger method?** The substantial empirical gains of `FlowOptimizer` can be attributed to three key factors.

1. **Expressiveness of flow-based models:** Flow-based

models provide a highly expressive parameterization of optimization updates, enabling the learned dynamics to represent complex transport maps that go well beyond fixed hand-designed update rules. This allows `FlowOptimizer` to adapt its behavior across different regions of the objective landscape and across iterations.

2. **Learning-to-optimize from problem geometry:** By training directly on optimization trajectories, `FlowOptimizer` learns recurrent patterns induced by the geometry of the underlying problem class, such as curvature, multi-modality, and scale, which are difficult to capture with generic optimizers. This enables the method to internalize when to favor exploration versus refinement in a data-driven manner.

3. **Learning population-level interactions:** Unlike pointwise update rules, `FlowOptimizer` operates on a population of candidate solutions and explicitly learns interactions among its members. This allows the population to coordinate exploration of the landscape, implicitly sharing information about promising regions, leading to more efficient behavior.

**Limitations.** One limitation of the proposed L2O architecture is the need to integrate the velocity field during the fine-tuning phase, which can lead to a long computational graph. While in all our experiments we employ a lightweight implementation based on Euler integration with only 5 integration steps, a promising direction for future work is to explore distillation schemes that would compress the integrated dynamics into a shallow surrogate model. Another limitation is that, in its current formulation, the proposed framework is designed for unconstrained optimization and can therefore incorporate constraints only indirectly via penalty or barrier terms. Extending flow-based L2O models to handle constraints, e.g., through projection operators, implicit layers, or integration with constrained optimization solvers, remains an important direction for future research.

## 6. Conclusion

We introduced **FlowOptimizer**, a learning-to-optimize framework that models optimization as an iterative, population-level transport process driven by a learned flow. By parameterizing each optimization step as a context-conditioned velocity field and training it through a combination of flow matching pre-training and deep unfolding with objective-driven fine-tuning, FlowOptimizer effectively integrates exploration and exploitation within a unified, self-supervised architecture. Across a wide range of non-convex benchmarks and real-world problems, the proposed method consistently outperforms classical sampling-based, gradient-based and learning-to-optimize baselines, while demonstrating strong generalization to higher-dimensional problems.

**Future Directions.** Several promising directions emerge from this work. Extending flow-based L2O architectures to constrained optimization in a principled manner, e.g., via projection operators or integration with constrained solvers, remains an important challenge. From a computational viewpoint, reducing the overhead induced by ODE integration through distillation could further improve efficiency.

## Acknowledgements

This work was partly supported by the National Science Foundation (NSF) CAREER Award #CCF-2238030.

## Impact Statement

This work introduces FlowOptimizer, a learning-to-optimize framework that leverages flow-based models and deep unfolding to construct data-driven optimization algorithms. By enabling scalable and amortized optimization for complex and high-dimensional problems, the proposed approach has the potential to reduce computational and engineering effort across a range of machine learning and control applications. As with other learned optimization methods, performance may degrade outside the training distribution, and caution is required in safety-critical settings where robustness and reliability are essential. This work does not involve sensitive data or pose direct societal risks and is intended to advance principled research on reliable learning-based optimization.

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

# A. Extended Related Work

**Sampling-Based Optimization.**   Sampling-based optimization algorithms rely on viewing optimization problems as the task of constructing and refining a proposal distribution to sample from low-cost regions of the objective landscape. This perspective has given rise to various fundamental algorithms such as the Cross-Entropy Method (CEM) (De Boer et al., 2005), which frames optimization as an iterative density estimation task. By minimizing the Kullback-Leibler (KL) divergence between a parameterized proposal typically Gaussian distribution and an implicit target distribution defined over a top-performing fraction of "elite" samples, CEM sequentially concentrates its probability mass around lower-cost regions. Nevertheless, CEM-type algorithms are known to suffer from premature mode collapse and convergence issues which limits its applicability, especially in high-dimensional non-convex settings. Similar algorithms such as stochastic search (Zhou & Hu, 2014; Wang et al., 2021), the model predictive path integral algorithm (Williams et al., 2017; Wang et al., 2022), etc. can also be found in various areas of engineering and machine learning.

A closely related family of methods are evolutionary strategies (ES), which similarly maintain and adapt a search distribution—typically Gaussian—but refine its parameters through stochastic, gradient-free perturbations weighted by sampled fitness. One of the most popular methods is the Covariance Matrix Adaptation Evolution Strategy (CMA-ES) (Hansen & Ostermeier, 2001; Hansen, 2016), a state-of-the-art black-box optimizer that adapts the full covariance of the proposal to capture local curvature and variable correlations, rendering it particularly effective on ill-conditioned, non-separable landscapes. More recent formulations such as Natural Evolution Strategies (NES) (Wierstra et al., 2014) recast these updates as natural-gradient ascent on the expected fitness, offering a principled information-geometric foundation (Ollivier et al., 2017) that unifies many ES and estimation-of-distribution algorithms. Despite their robustness, these approaches rely on hand-crafted, myopic update rules and a restrictive Gaussian parameterization, motivating learning-based proposals capable of representing richer, multimodal distributions and algorithmic updates.

**Particle Swarm Optimization.**   Particle swarm optimization (PSO) is a population-based, derivative-free metaheuristic in which candidate solutions explore the search space via velocities that blend each particle's own best location with the swarm's best (Kennedy & Eberhart, 1995), later refined with inertia weights (Shi et al., 1998), constriction coefficients (Clerc & Kennedy, 2002), and the Standard PSO 2011 variant (Zambrano-Bigiarini et al., 2013). Its update rule, however, is a hand-designed linear combination of cognitive and social terms with hand-tuned coefficients. FlowOptimizer can be seen as a learned generalization: it likewise evolves a population through a velocity field, but replaces PSO's fixed heuristic with a learned velocity field, adapting the population's trajectories to the problem family rather than relying on static swarm dynamics.

**Gradient-Based Non-Convex Optimization.**   Gradient-based optimization methods rely fundamentally on local derivative information to guide their updates (Wright et al., 1999). Although their theoretical foundations and convergence guarantees are rooted in the convex optimization setting (Boyd & Vandenberghe, 2004), gradient-based methods are very commonly used for navigating the complex, high-dimensional non-convex loss surfaces. In practice, the workhorse of large-scale learning is stochastic gradient descent (SGD) (Robbins & Monro, 1951; Bottou et al., 2018), which substitutes the full gradient with cheap, noisy minibatch estimates. To improve conditioning and alleviate the burden of manual step-size tuning, a family of adaptive methods rescales updates using accumulated gradient statistics, including AdaGrad (Duchi et al., 2011), which adapts per-coordinate learning rates from the running sum of squared gradients, and Adam (Kingma & Ba, 2015), which couples momentum with an exponentially decaying second-moment estimate and has become the de facto optimizer for training deep networks. Beyond such first-order schemes, second-order methods incorporate curvature information for faster local convergence: Newton's method rescales the gradient by the inverse Hessian (Wright et al., 1999), quasi-Newton schemes such as BFGS and its limited-memory variant L-BFGS (Liu & Nocedal, 1989) build low-rank Hessian approximations from successive gradients, and structured variants such as Differential Dynamic Programming (Jacobson & Mayne, 1970) exploit problem-specific structure (e.g., the temporal structure of dynamics in trajectory optimization) have been shown to be quite effective in large-scale non-convex optimization settings (Saravanos et al., 2023). Yet all such methods remain fundamentally local, relying on derivative information that offers little guidance on the rugged, high-dimensional non-convex landscapes encountered in practice.

**Learning-to-Optimize and Deep Unfolding.**   Learning-to-optimize methodologies aim to learn optimization algorithms by training on problem data (Chen et al., 2022; Shlezinger et al., 2022; Amos et al., 2023). A first class of approaches learns *optimization proxies* that map problem instances directly to near-optimal solutions (Amos et al., 2023); these are especially effective for parametric problems that must be solved repeatedly, where supervised and self-supervised primal-dual

schemes recover high-quality feasible solutions orders of magnitude faster than classical solvers (Fioretto et al., 2020; Park & Van Hentenryck, 2023; Van Hentenryck, 2025). A second, more structured class is *deep unfolding*, which unrolls iterative optimization algorithms as sequential layers of a neural network, enabling end-to-end learning of algorithmic parameters (Monga et al., 2021; Zhang et al., 2020). Recent works further advance this paradigm through more expressive architectures and learning strategies (Sambharya & Stellato, 2024; Saravanos et al., 2025; Oshin et al., 2026), improving performance and generalization across diverse tasks. A third, complementary line treats optimization itself as a *differentiable* component—embedding parametric programs as implicit layers and enforcing hard constraints through differentiable feasibility steps (Amos & Kolter, 2017; Donti et al., 2017; 2021; Nguyen & Donti, 2026).

**Generative Modeling.** Generative models learn to transport a simple reference distribution to a complex data distribution. *Diffusion models* cast this as the time reversal of a noising process, training a network to denoise or estimate the score of the noised marginals (Ho et al., 2020; Song & Ermon, 2019), with a continuous-time SDE formulation and an equivalent probability-flow ODE (Song et al., 2021). *Flow-based* models instead learn a velocity field whose ODE carries the reference to the target (Chen et al., 2018), a view closely tied to optimal transport and Wasserstein gradient flows (Xie & Cheng, 2026). A growing body of work repurposes such generative models as optimization processes, recasting the search for high-quality solutions as sampling from a target distribution concentrated on low-cost regions of the objective. Their appeal lies in representing rich, multimodal solution families far beyond the unimodal Gaussians of classical sampling-based methods, though their use in end-to-end optimization is complicated by the implicit nature of the target and the lack of ground-truth solutions. Recent efforts couple generative sampling with feasibility or control structure (Idoko & Singh, 2025; Kurtz & Burdick, 2025), frame optimization or constrained transport as reward-tilted, KL-regularized, or marginal-constrained sampling solved via stochastic optimal control (Domingo-Enrich et al., 2025; So et al., 2026; Theodoropoulos et al., 2025; 2026), and more broadly seek to bridge generative modeling and optimization across discrete and continuous domains (Uehara et al., 2024; Kong et al., 2025; Mazé & Ahmed, 2023; Xie & Cheng, 2026). The field nonetheless remains nascent, with most approaches still facing scalability and expressiveness limitations.

**Meta-Black-Box Optimization.** A parallel line of L2O research is Meta-Black-Box Optimization (MetaBBO), which uses meta-learning to automatically design or configure optimizers that generalize across a distribution of problems (Ma et al., 2025). MetaBBO typically adopts a bi-level structure: a meta-level policy dictates the algorithm selection, hyperparameter configuration, update rules, or solution manipulation of a low-level black-box optimizer, often a population-based evolutionary algorithm (Ma et al., 2025; Chen et al., 2024; Li et al., 2025). The defining assumption of this setting is that objectives are *non-differentiable* and expensive to query, so performance is measured under a function-evaluation budget and the low-level optimizer is derivative-free. FlowOptimizer occupies the complementary regime: it assumes access to differentiable objectives and targets low wall-clock cost rather than evaluation count, learning a velocity field by flow matching and refining it through deep unfolding.

# B. Experiment Details

## B.1. General Information

For all algorithms, we use a population size $P = 20$. In the FlowOptimizer architecture, the Gaussian base sampler was used for the Rastrigin problems, as it was observed to facilitate exploration. For all other problems, the previous population was directly fed into the velocity field model. All experiments were performed on an RTX 4090 GPU 24GB.

## B.2. Architecture Details

In all experiments, the velocity field is parameterized by a permutation-equivariant neural network with hidden dimension $h = 128$. Each population member $z_t^{(i)} \in \mathbb{R}^n$ is first embedded using a linear layer $\mathbb{R}^n \to \mathbb{R}^{128}$. The per-member context features are concatenated and embedded using a separate linear layer to $\mathbb{R}^{128}$. The state and context embeddings are concatenated to form a 256-dimensional representation per population member. Population interactions are modeled using a single multi-head self-attention layer with 4 heads and embedding dimension 256, followed by a residual connection and layer normalization. The resulting per-member features are concatenated with global conditioning features shared across the population, including the scalar flow time, optional averaged gradient information, and summary statistics from a history buffer of length $H = 3$. History features consist of the mean and standard deviation of past populations. The final per-member velocity is produced by an MLP with 3 hidden layers of width 128, each followed by layer normalization and ELU activations, and a final linear layer mapping to $\mathbb{R}^n$.

## B.3. Training Details

For all problem classes, we use 1000 problems for training, 200 for validation and 200 held-out ones for testing. The batch size is set to 50. The Adam optimizer with learning rate $10^{-4}$ is used in all experiments.

**Pre-training via flow matching loss.**    In the pre-training phase, we construct the flow matching dataset pairs as follows. We sample a base population of $P$ points via a Gaussian distribution with same mean and covariance as the initializations $\mu^0$ and $\Sigma^0$ for each problem class. Then, the improvement procedure takes place through sampling $50P$ points with a Gaussian that has the same mean and three times larger covariance. Out of the obtained samples, we select the best $P$ out of them and set them as $\mathcal{X}^+$.

**Objective-driven end-to-end fine-tuning.**    During the fine-tuning phase, the balancing parameter in the training loss is set to $\alpha = 0.8$. Initially, we follow a progressive training scheme where each iteration-specific flow model is trained using the sum of per-iteration losses up to that iteration, while the others are frozen. At each cycle each iteration-specific model is trained for 10 epochs except for the last one which is trained for 60 epochs. This process takes place for 1800 epochs. During the last 200 epochs all models are trained simultaneously using the total loss across iterations.

## B.4. Baselines

**Random Search (RS).**    Random Search (RS) samples candidate solutions from a fixed distribution and returns the one with the lowest objective value. Across all problems, RS samples from a uniform distribution supported on the entire domain of the function. RS serves as a fundamental naive baseline, included to verify that the gains of more sophisticated methods stem from exploiting problem structure rather than from sample coverage alone.

**Cross-Entropy Method (CEM).**    The Cross-Entropy Method (CEM) is a sampling-based optimization algorithm that iteratively updates a parametric sampling distribution to concentrate probability mass on regions of the search space that correspond to low objective values. At iteration $k$, a population of $P$ samples $\{x_j^k\}_{j=1}^P$ is drawn from a Gaussian distribution $x_j^k \sim \mathcal{N}(\mu^k, \Sigma^k)$, and then each sample is evaluated to obtain the values $f(x_j^k)$. Then, the next step is to update the distribution parameters $\mu^{k+1}, \Sigma^{k+1}$ to be used in the next iteration. In the most common "hard" variant of CEM, an elite set $\mathcal{E}^k \subset \{1, \ldots, P\}$ consisting of the $P_e$ samples with the lowest objective values is selected. The distribution parameters are then updated by maximum likelihood estimation over the elite samples as follows

$$\mu^{k+1} = \frac{1}{P_e} \sum_{j \in \mathcal{E}^k} x_j^k, \qquad \Sigma^{k+1} = \frac{1}{P_e} \sum_{j \in \mathcal{E}^k} (x_j^k - \mu^{k+1})(x_j^k - \mu^{k+1})^\top. \tag{14}$$

The elite fraction is set to the reasonable default $P_e/P = 0.2$.

**Covariance Matrix Adaptation Evolution Strategy (CMA-ES).**    CMA-ES is a population-based evolutionary strategy that adapts a Gaussian search distribution by updating its mean, covariance matrix and global step size. At iteration $k$, a population of $P$ samples is generated as

$$x_j^k = \mu^k + \sigma^k y_j^k, \qquad y_j^k \sim \mathcal{N}(0, \Sigma^k), \quad j = 1, \ldots, P, \tag{15}$$

where $\mu^k \in \mathbb{R}^n$ is the mean, $\Sigma^k \in \mathbb{R}^{n \times n}$ is the covariance matrix, and $\sigma^k > 0$ is a global step-size parameter.

The samples are evaluated and ranked according to their objective values. Let $\{x_{(j)}^k\}_{j=1}^{P_e}$ denote the $P_e$ best-performing samples. The mean is updated via weighted recombination:

$$\mu^{k+1} = \sum_{j=1}^{P_e} w_j x_{(j)}^k, \qquad \sum_{j=1}^{P_e} w_j = 1, \ w_j > 0. \tag{16}$$

The covariance matrix is updated to reflect the directions of successful search steps:

$$\Sigma^{k+1} = (1 - c_\mu)\Sigma^k + c_\mu \sum_{j=1}^{P_e} w_j \frac{(x_{(j)}^k - \mu^k)(x_{(j)}^k - \mu^k)^\top}{(\sigma^k)^2}, \tag{17}$$

where $c_\mu \in (0, 1)$ is a learning rate. The global step size $\sigma^k$ is adapted using cumulative step-size adaptation, which increases $\sigma^k$ when consecutive steps are aligned and decreases it otherwise.

**Particle Swarm Optimization (PSO).** Particle Swarm Optimization (PSO) is a population-based metaheuristic in which a swarm of $P$ particles explores the search space by combining each particle's own experience with that of the swarm. We use the standard SPSO-2011 configuration (**?**). As with the other algorithms, the initial particle positions $\{x_j^0\}_{j=1}^P$ are drawn from a Gaussian $\mathcal{N}(\mu, \Sigma)$ matching the initialization used by the other algorithms for the same problem class, and the velocities $\{v_j^0\}_{j=1}^P$ are initialized to zero. Each particle $j$ maintains its personal best position $p_j^k$, i.e. the lowest-objective point it has visited up to iteration $k$, and the swarm maintains the global best position $g^k$ over all particles. At iteration $k$, the velocity and position of each particle are updated as

$$v_j^{k+1} = w\, v_j^k + c_1\, r_1 \odot \left(p_j^k - x_j^k\right) + c_2\, r_2 \odot \left(g^k - x_j^k\right), \tag{18}$$

$$x_j^{k+1} = x_j^k + v_j^{k+1}, \qquad j = 1, \ldots, P, \tag{19}$$

where $\odot$ denotes elementwise multiplication, $r_1, r_2 \sim \mathcal{U}([0,1]^n)$ are sampled independently at every step, $w$ is the inertia weight, and $c_1, c_2$ are the cognitive and social coefficients respectively. We adopt the SPSO-2011 defaults $w = 1/(2\ln 2) \approx 0.7213$ and $c_1 = c_2 = 1/2 + \ln 2 \approx 1.1931$, which are not tuned. The algorithm returns the global best position found across all particles and iterations.

**Multi-Start Gradient Descent (MS-GD).** Multi-Start Gradient Descent (MS-GD) is a simple strategy that mitigates sensitivity to initialization in gradient-based optimization by running multiple independent gradient descent trajectories from different starting points. Specifically, we draw $P$ initial points $\{x_j^0\}_{j=1}^P$ from a Gaussian distribution $\mathcal{N}(\mu, \Sigma)$, matching the initialization used by the other algorithms for the same problem class. Each initialization is then evolved independently using gradient descent updates of the form

$$x_j^{k+1} = x_j^k - \eta \nabla f(x_j^k), \qquad j = 1, \ldots, P, \tag{20}$$

where $\eta > 0$ is a fixed step size. The algorithm returns the best solution across all trajectories. For tuning, we perform a grid search over $\eta \in \{10^{-3}, 2 \cdot 10^{-3}, 5 \cdot 10^{-3}, \ldots, 1.0, 2.0, 5.0\}$.

**Multi-Start Nesterov Accelerated Gradient Descent (MS-NAG).** Multi-Start Nesterov Accelerated Gradient Descent (MSNAG) extends MS-GD by incorporating acceleration into each trajectory. As in MS-GD, we draw $P$ initial points $\{x_j^0\}_{j=1}^P$ from a Gaussian distribution $\mathcal{N}(\mu, \Sigma)$, matching the initialization used by the other algorithms for the same problem class. Each trajectory is then evolved independently according to the Nesterov accelerated updates

$$y_j^k = x_j^k + \beta \left(x_j^k - x_j^{k-1}\right), \tag{21}$$

$$x_j^{k+1} = y_j^k - \eta \nabla f(y_j^k), \qquad j = 1, \ldots, P, \tag{22}$$

where $\eta > 0$ is a fixed step size and $\beta \in [0, 1)$ is the momentum coefficient. The algorithm returns the best solution across all trajectories. We set the momentum coefficient $\beta = 0.9$ and tune the step size $\eta$ through performing a grid search over $\eta \in \{10^{-3}, 2 \cdot 10^{-3}, 5 \cdot 10^{-3}, \ldots, 1.0, 2.0, 5.0\}$.

**Multi-Start Adam (MS-Adam).** Multi-Start Adam (MS-Adam) extends MS-GD by replacing the plain gradient update with the Adam adaptive moment estimation scheme. As before, we draw $P$ initial points $\{x_j^0\}_{j=1}^P$ from a Gaussian distribution $\mathcal{N}(\mu, \Sigma)$, matching the initialization used by the other algorithms for the same problem class. Each trajectory maintains first- and second-moment estimates $m_j^k, v_j^k$ (initialized at zero) and is evolved independently according to the Adam updates

$$g_j^k = \nabla f(x_j^k), \tag{23}$$

$$m_j^{k+1} = \beta_1 m_j^k + (1 - \beta_1) g_j^k, \tag{24}$$

$$v_j^{k+1} = \beta_2 v_j^k + (1 - \beta_2) \left(g_j^k\right)^2, \tag{25}$$

$$\hat{m}_j^{k+1} = \frac{m_j^{k+1}}{1 - \beta_1^{k+1}}, \qquad \hat{v}_j^{k+1} = \frac{v_j^{k+1}}{1 - \beta_2^{k+1}}, \tag{26}$$

$$x_j^{k+1} = x_j^k - \eta\, \frac{\hat{m}_j^{k+1}}{\sqrt{\hat{v}_j^{k+1}} + \epsilon}, \qquad j = 1, \ldots, P, \tag{27}$$

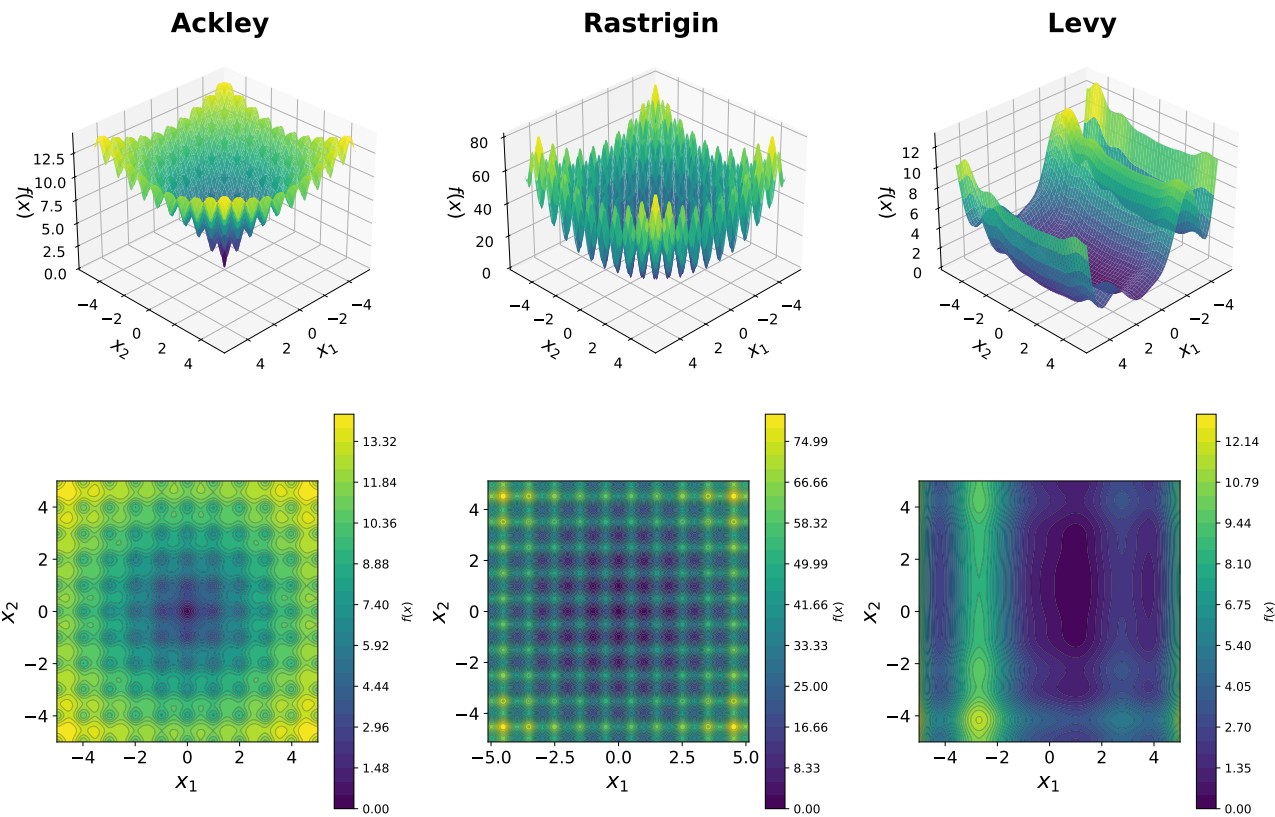

*Figure 6.* **2D Illustration of Ackley, Rastrigin and Levy Functions.** These optimization benchmarks are highly non-convex, with their complexity further growing as dimension increases.

where $\alpha > 0$ is the learning rate, all operations are applied elementwise, and $\epsilon > 0$ is a small constant for numerical stability (fixed to $10^{-8}$). The moment decay rates are fixed to their standard values $\beta_1 = 0.9$ and $\beta_2 = 0.999$. The algorithm returns the best solution across all trajectories. The learning rate $\eta$ is set to the standard default $\eta = 10^{-3}$.

**Multi-Start Newton (MS-Newton).**   Multi-Start Newton (MS-Newton) extends MS-GD by incorporating second-order curvature information into each trajectory. As before, we draw $P$ initial points $\{x_j^0\}_{j=1}^P$ from a Gaussian distribution $\mathcal{N}(\mu, \Sigma)$, matching the initialization used by the other algorithms for the same problem class. Each trajectory is then evolved independently according to the damped Newton updates

$$x_j^{k+1} = x_j^k - \left(\nabla^2 f(x_j^k) + \lambda I\right)^{-1} \nabla f(x_j^k), \qquad j = 1, \ldots, P, \tag{28}$$

where $\lambda \geq 0$ is a damping (Levenberg–Marquardt) parameter that regularizes the Hessian, ensuring the linear system is well-conditioned and the step remains a descent direction when $\nabla^2 f(x_j^k)$ is indefinite, as is common for the nonconvex problem classes considered here. The algorithm returns the best solution across all trajectories. The damping parameter is set to a reasonable default $\lambda = 10^{-3}$.

**Multi-Start Variant of Learning-to-Optimize Gradient Descent (L2O-GD).**   We also consider a L2O variant of MSGD where we learn the "optimal" stepsizes for every iteration of the gradient descent updates as in (Sambharya & Stellato, 2024). During inference, we draw $P$ samples from the same initial distributions as the rest of the algorithms, and apply the same learned updates on all samples. At the end, the algorithm returns the best solution across all trajectories.

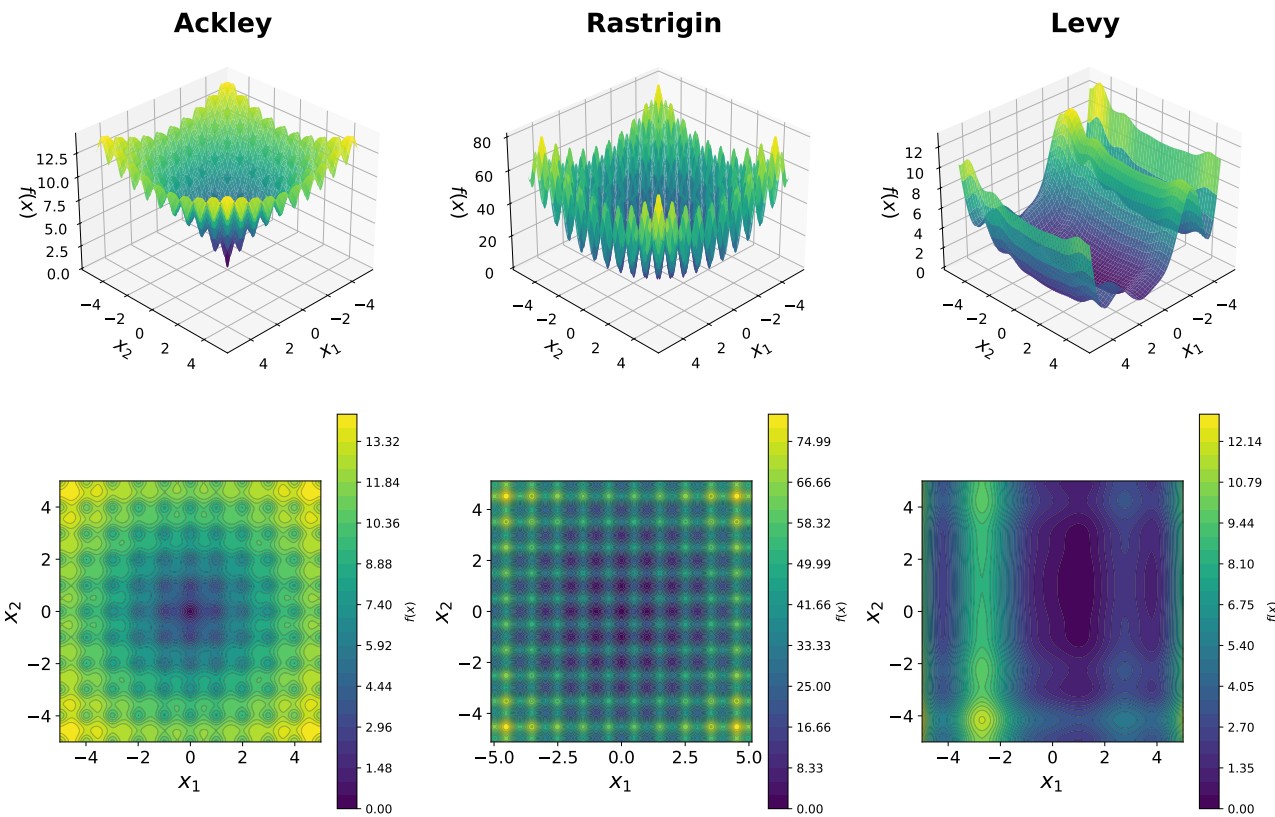

*Figure 7.* **2D Illustration of Ackley, Rastrigin and Levy Functions.** These optimization benchmarks are highly non-convex, with their complexity further growing as dimension increases.

### B.5. Standard Optimization Benchmarks

Here, we provide a detailed description of the non-convex optimization benchmark functions used in our experiments, along with their configurations. For comprehensive references on such functions, the reader is referred to (Molga & Smutnicki, 2005; Naser et al., 2025). A 2D visualization of the functions used (Ackley, Rastrigin and Levy) is provided in Fig. 7.

**Ackley function.** The Ackley function is given by

$$f(x) = -a \exp\left(-b\sqrt{\frac{1}{n}\sum_{i=1}^{n}(x_i - c_i)^2}\right) - \exp\left(\frac{1}{n}\sum_{i=1}^{n}\cos(\omega(x_i - c_i))\right) + a + \exp(1), \tag{29}$$

where the parameter $a > 0$ controls the amplitude of the outer exponential, $b > 0$ governs the decay rate of the first exponential term, $\omega > 0$ is the cosine frequency, controlling the density of local minima and $c = [c_1, \ldots, c_n] \in \mathbb{R}^n$ is a shift vector. The ground-truth global optimum of this function is found at $x^* = c$ with $f^* = 0$.

Across our experiments, we consider the domain to be $x \in [-30, 30]^n$, and the problem parameters to be randomly drawn from $a \sim \text{LogUniform}[10, 40]$, $b \sim \text{LogUniform}[0.1, 0.4]$, $\omega \sim \text{Uniform}[\pi, 2\pi]$ and $c \sim \text{Uniform}[-3, 3]^n$. For all algorithms, the particles of the initial population are randomly drawn from $x_0^j \sim \mathcal{N}(\mu_0, \Sigma_0)$, $j = 1, \ldots, P$, where $\mu_0 \sim \text{Uniform}[-25, 25]^n$ and $\Sigma_0 = I_n$.

**Rastrigin function.** The Rastrigin function is given by

$$f(x) = an + \sum_{i=1}^{n} \left( (x_i - c_i)^2 - a \cos \left( 2\pi(x_i - c_i) \right) \right), \tag{30}$$

where the parameter $a > 0$ controls the amplitude of the cosine term and $c = [c_1, \ldots, c_n] \in \mathbb{R}^n$ is a shift vector. The ground-truth global optimum is found at $x^* = c$ with $f^* = 0$.

Across our experiments, we consider the domain to be $x \in [-10, 10]^n$, and the problem parameters are randomly drawn from $a \sim \text{Uniform}[0, 5]$ and $c \sim \text{Uniform}[-3, 3]^n$. For all algorithms, the particles of the initial population are randomly drawn from $x_0^j \sim \mathcal{N}(\mu_0, \Sigma_0)$, $j = 1, \ldots, P$, where $\mu_0 \sim \text{Uniform}[-8, 8]^n$ and $\Sigma_0 = I_n$.

**Levy function.** The Levy function is given by

$$f(x) = \sin^2(\pi w_1) + \sum_{i=1}^{n-1} (w_i - 1)^2 \left( 1 + 10 \sin^2(\pi w_i + 1) \right) + (w_n - 1)^2 \left( 1 + \sin^2(2\pi w_n) \right), \tag{31}$$

where

$$w_i = 1 + \frac{(x_i - c_i)/\alpha - 1}{4}, \qquad i = 1, \ldots, n, \tag{32}$$

where the parameter $\alpha > 0$ is a scaling factor that stretches or compresses the input space, and $c = [c_1, \ldots, c_n] \in \mathbb{R}^n$ is a shift vector. The ground-truth global optimum is located at $x^* = c$ with $f^* = 0$.

Across our experiments, we consider the domain $x \in [-10, 10]^n$, and the problem parameters are randomly drawn from $\alpha \sim \text{LogUniform}[0.2, 5]$ and $c \sim \text{Uniform}[-3, 3]^n$. For all algorithms, the particles of the initial population are randomly drawn from $x_0^j \sim \mathcal{N}(\mu_0, \Sigma_0)$, $j = 1, \ldots, P$, where $\mu_0 \sim \text{Uniform}[-8, 8]^n$ and $\Sigma_0 = I_n$.

### B.6. Real-World Application Problems

**Power Grid Economic Load Dispatch.** We consider the Economic Load Dispatch (ELD) problem in power systems, a classical non-convex optimization problem that aims to determine the power output of multiple generators so as to minimize total generation cost while satisfying demand constraints (Sinha et al., 2003). The objective function is given by

$$f(x) = \sum_{i=1}^{N_g} \left[ a_i x_i^2 + b_i x_i + c_i + \left| d_i \sin(e_i(P_i^{\min} - x_i)) \right| \right] + \lambda \left( \sum_{i=1}^{N_g} x_i - P_D \right)^2 \tag{33}$$

where $N_g$ is the number of generators and $x = [x_1; \ldots; x_{N_g}] \in \mathbb{R}^{N_g}$ is the vector of power outputs. Each generator has the fuel cost coefficients $a_i, b_i, c_i$, the valve-point loading coefficients $d_i, e_i$ and a minimum generation limit $P_i^{\min}$. In addition, the generators need to meet the total power demand $P_D$.

**Supply Chain Multi-Source Weber Problem.** We consider the Multi-Source Weber Problem (MSWP), a classical facility location problem arising in supply chain and logistics applications (Brimberg et al., 2008). The objective is to determine the locations of $K$ facilities so as to minimize the total weighted distance to a set of $M$ customers, and is given by

$$f(X) = \sum_{j=1}^{M} w_j \min_{k=1\ldots K} \|x_k - a_j\|_2 \tag{34}$$

where $X = \{x_1, \ldots, x_K\}$ are the coordinates of the $K$ facilities (optimization variables), $\mathbf{a}_j$ are the fixed locations of the $M$ customers, $w_j$ is the demand weight of customer $j$ (usually set to 1.0).

We consider the smoothed objective using the SoftMin operator as follows:

$$f(X) = \sum_{j=1}^{M} w_j \left( -\frac{1}{\gamma} \log \sum_{k=1}^{K} \exp \left( -\gamma \|x_k - a_j\|_2 \right) \right) \tag{35}$$

where $\gamma > 0$ is a temperature parameter controlling the smoothness of the approximation. As $\gamma \to \infty$, the SoftMin function approaches the hard minimum, $f_\gamma(X) \to f(X)$, recovering the original non-smooth objective. This relaxation renders the landscape differentiable everywhere, enabling the application of gradient-based baselines such as Nesterov momentum and Adam, while preserving the non-convex structure inherent to the facility location problem.

**Robotic Arm Inverse Kinematics.** We also consider an inverse kinematics task involving a planar robot arm with $D$ degrees of freedom. The decision variable is the vector of joint angles $\Theta \in [-\pi, \pi]^D$, and the objective is to find a configuration such that the end-effector position $p_{\text{ee}}(\Theta)$ coincides with a target location $p_{\text{target}}$ while avoiding collisions with a set of $M$ circular obstacles. The full objective is defined as a weighted sum of a target-reaching term and a collision-avoidance penalty,

$$f(\Theta) = \|p_{\text{ee}}(\Theta) - p_{\text{target}}\|_2^2 + \lambda \sum_{i=1}^{N} \sum_{j=1}^{M} \text{ReLU}(R_j - \|s_i(\Theta) - o_j\|_2)^2 , \tag{36}$$

where $p_{\text{ee}}(\Theta)$ is obtained via the forward kinematics chain, and the second term penalizes penetrations into obstacles by discretizing the arm into $N$ sample points $s_i(\Theta)$ located along the links. Each obstacle is modeled by a center $o_j$ and a radius $R_j$, and the ReLU operator ensures that the penalty is active only when a sample point lies inside an obstacle. The scalar $\lambda$ balances the target-reaching and collision-avoidance objectives. This optimization landscape is highly non-convex and multimodal, as feasible solutions often require navigating around obstacles, forcing the optimizer to escape local minima corresponding to configurations that are close to the target but infeasible due to collisions.

## C. Additional Experiments

### C.1. 1D Flat Middle Region Example with Identical Initializations

In this example, we evaluate the ability of FlowOptimizer to navigate towards the optimum in cases where different problem instances share an identical initial population located in a flat region of the objective landscape. This is a quite interesting example, as it studies how our flow-based model effectively explores the landscape despite its deterministic nature. We consider the following function:

$$f(x) = \underbrace{\frac{1 - \text{sgn}(\mu)\,\text{sgn}(x)}{2}}_{\text{bad-side mask}} \Big[\kappa x^2 + B, \varepsilon(x)\sin(\omega x)\Big] - A\exp\left(-\frac{(x-\mu)^2}{2\sigma^2}\right) \tag{37}$$

where $\varepsilon(x) = 1 - (1 + (x/W)^8)^{-1}$, with fixed $\kappa = 0.05$, $B = 0.12$, $\omega = 2.8$, $W = 1.6$, and per-instance parameters drawn uniformly as $\mu \sim \mathcal{U}(2.5, , 4.0)$ (sign randomized), $A \sim \mathcal{U}(1.2, , 2.8)$, $\sigma \sim \mathcal{U}(0.25, 0.40)$. The domain is $[-5.5, , 5.5]$ and populations are initialized with mean $\sim \mathcal{U}(-0.5, 0.5)$ and std 0.2, placing them squarely in the flat middle region.

We consider five distinct objective functions that share the same initialization (Fig. 8). As expected, the first iteration produces identical population updates across all instances, since both the population and the available historical information are identical. However, the learned update policy has acquired the ability to actively spread particles when operating in flat regions. Consequently, small differences in objective evaluations become amplified over subsequent iterations, leading to increasingly distinct population trajectories across problem instances. By iteration 4, the populations have already diverged substantially, allowing the optimizer to identify promising regions of the search space. As shown in Fig. 8, the trajectories subsequently converge toward the correct optima of their respective objective functions.

This experiment illustrates that identical initial populations do not imply identical optimization trajectories beyond the first iteration. The incorporation of historical context enables FlowOptimizer to progressively disambiguate problem instances and adapt its search behavior accordingly.

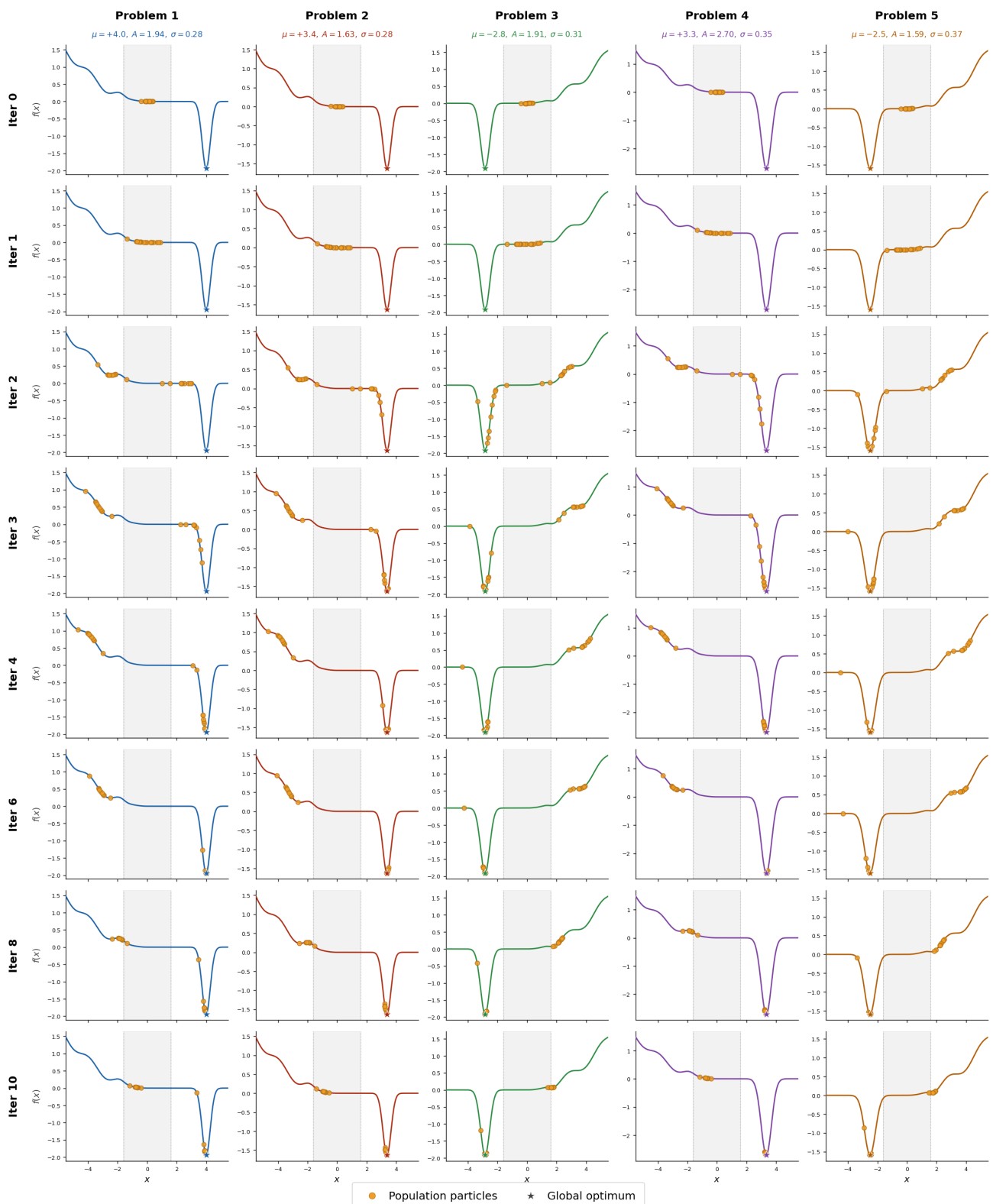

*Figure 8.* **1D Flat Middle Region Example with Identical Initial Populations.**

## C.2. Detailed 2D Ackley Example

Next, we demonstrate a 2D Ackley example to clearly illustrate the underlying velocity fields the FlowOptimizer learns as well as the trajectories of the populations. In Figs. 9 and 10, the learned velocity fields and population iterates of our methodology are illustrated across different iterations. Notably, as the iterations progress, the velocity field tends to point towards the global optimum despite the very high number of local minima in the landscape.

In Fig. 11, we see the samples trajectories for all methods for a specific problem instance, while Figs. 12 and 13 show the samples of each method across iterations. This illustration highlights the ability of the FlowOptimizer methodology to drive the points towards the optimum fast in contrast with the considered baselines.

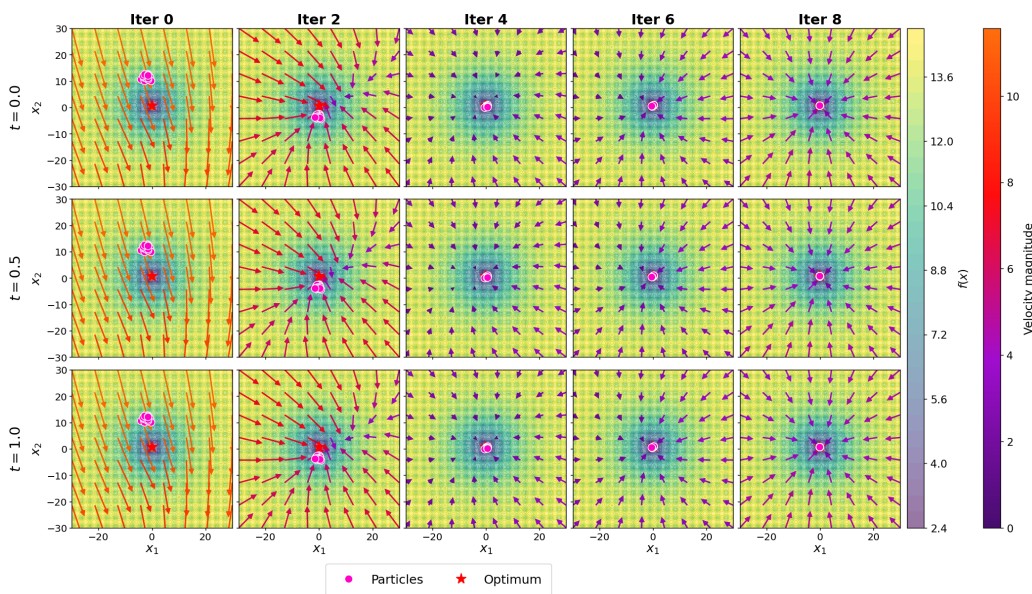

*Figure 9.* **Visualization of FlowOptimizer Velocity Field - 2D Ackley Problem - Full Domain.**

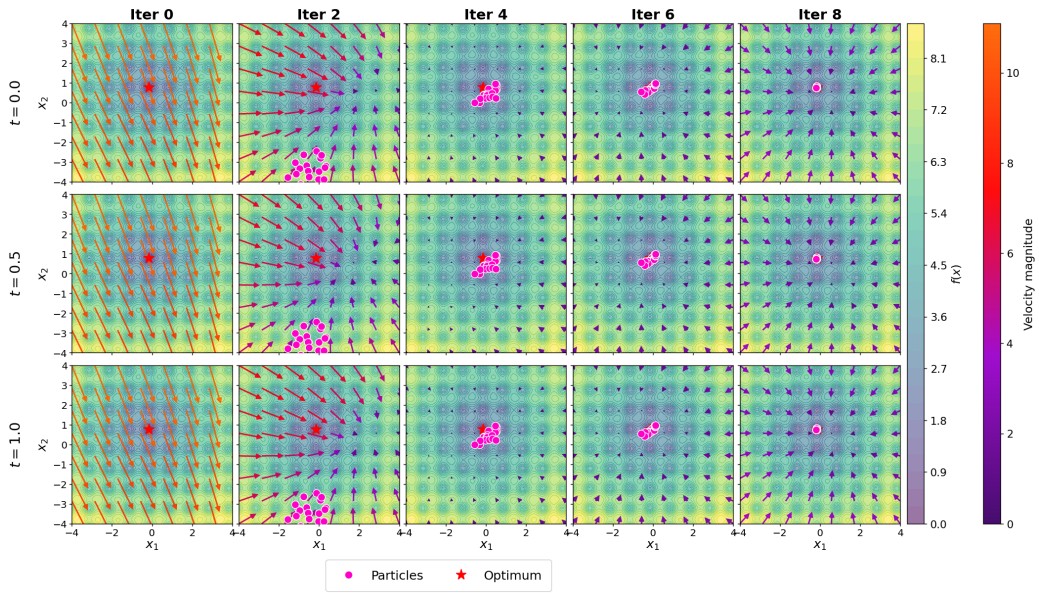

*Figure 10.* **Visualization of FlowOptimizer Velocity Field - 2D Ackley Problem - Zoomed into $[-4, 4]^2$.**

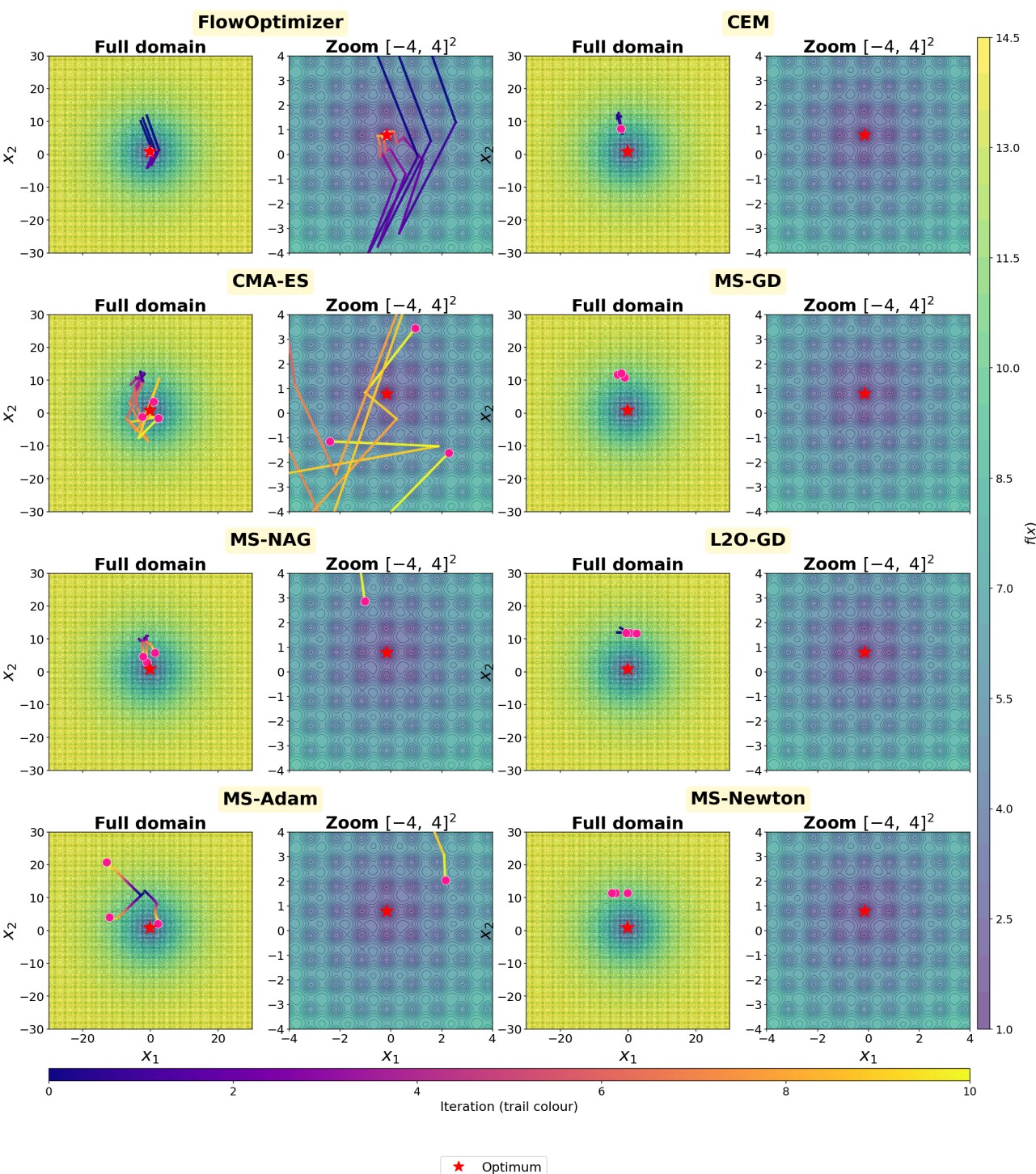

*Figure 11.* **Visualization of Trajectories of Three Selected Particles Per Method - 2D Ackley Problem.**

**2D Visualization of Particles Trajectories for All Algorithms (Ackley)**

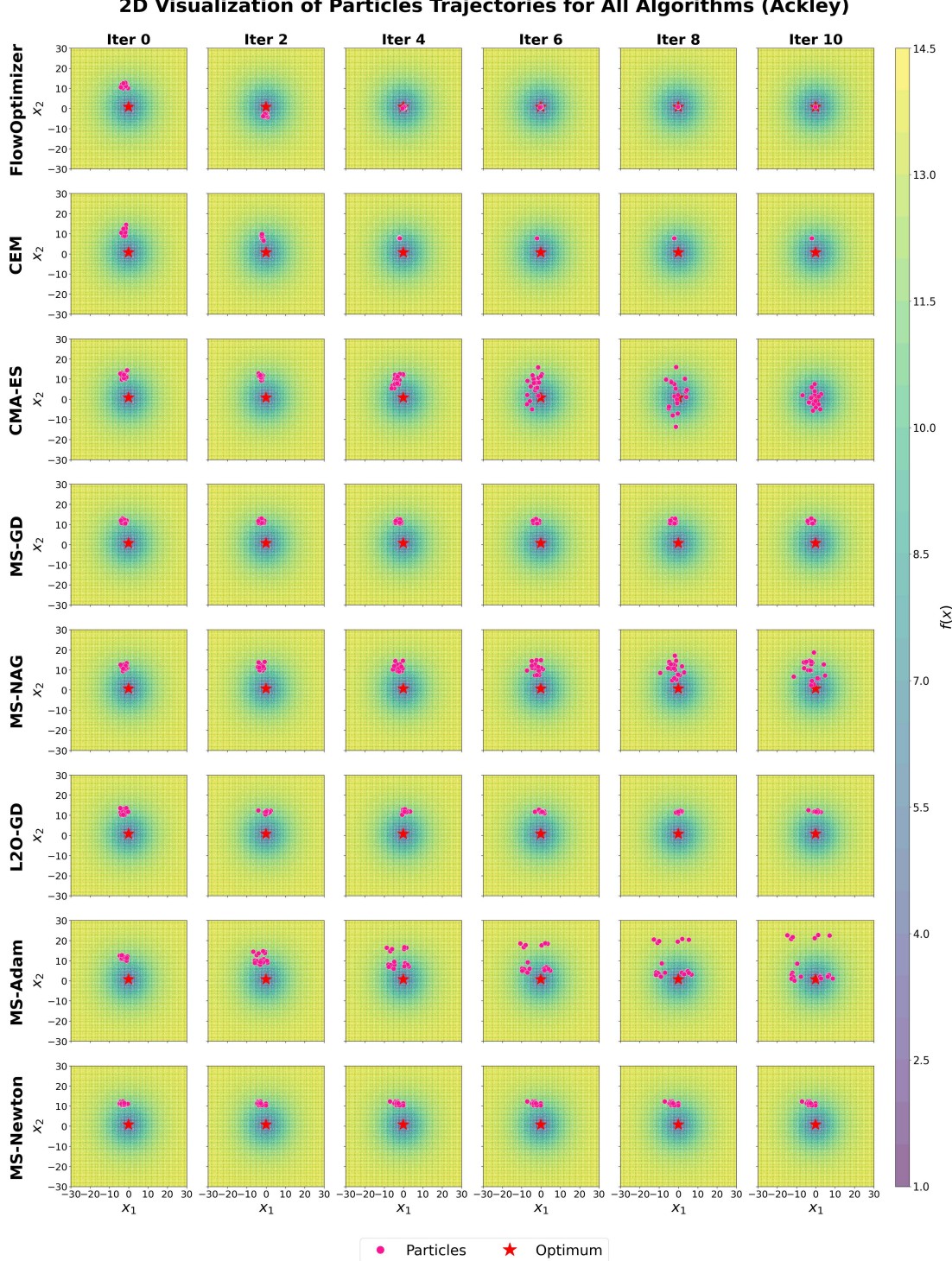

*Figure 12.* **Visualization of Iteration Snapshots for All Methods - 2D Ackley Problem - Full Domain.**

## 2D Visualization of Particles Trajectories for All Algorithms (Ackley)

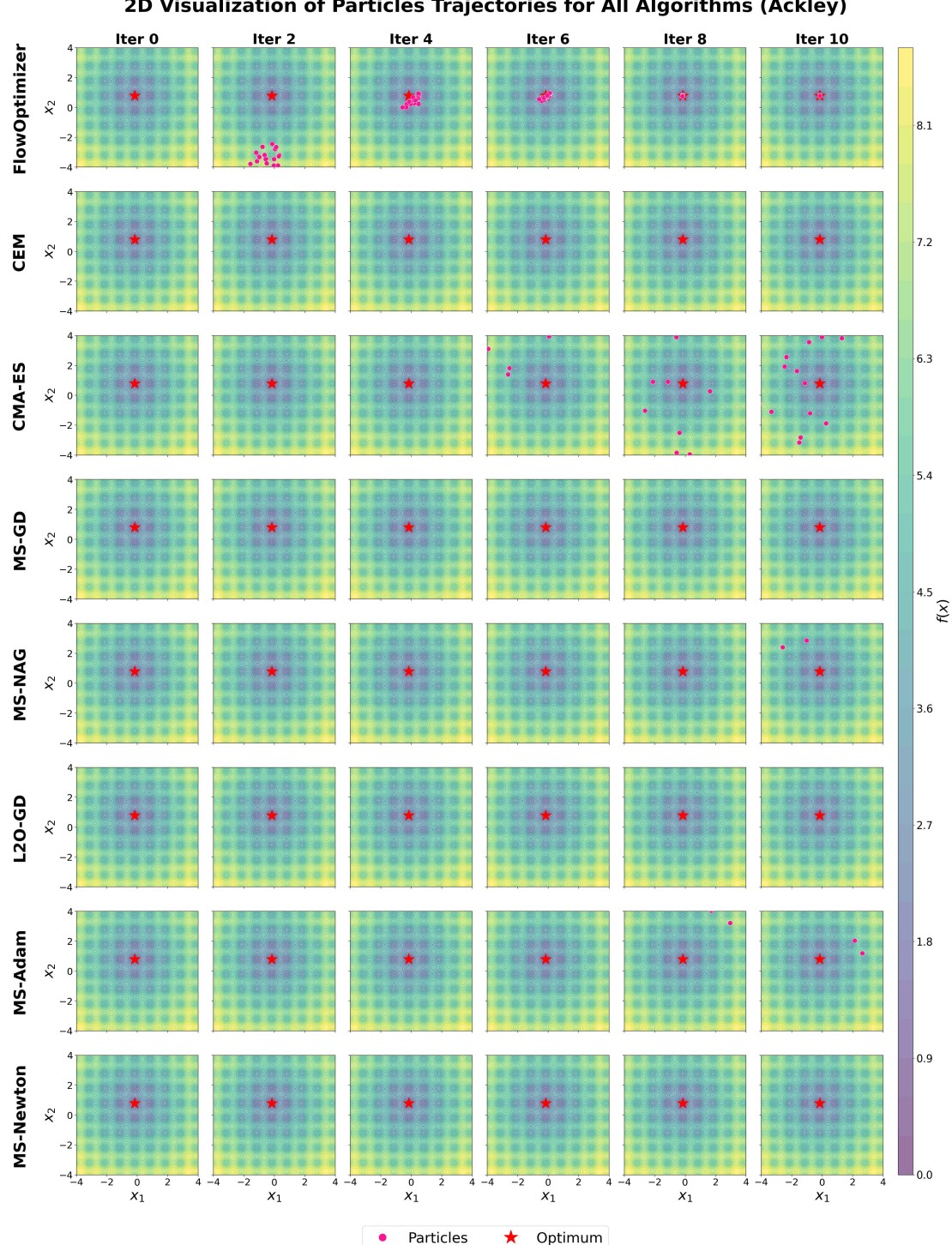

*Figure 13.* **Visualization of Iteration Snapshots for All Methods - 2D Ackley Problem - Zoomed into** $[-4, 4]^2$**.**

## C.3. Ablation Studies

Next, we provide several ablation studies. In Table 2, we vary the parameter $\alpha$. Our observations show that the best performing values are between $0.8$ and $0.9$ striking an optimal balance between prioritizing the best point of the population and the mean. Table 3 provides an ablation on the population size for the 20D standard optimization benchmarks. The results show that above some critical population size, the FlowOptimizer can learn to navigate to high-quality solutions. Table 4 provides a comparison between using the Gaussian base sampler or not. Although our observations show that in general it is preferable to feed the population directly to the new iteration, we find that in some cases the additional sampling step might benefit exploration. In Table 5, we compare the performance of our method after pre-training and after fine-tuning across various dimensions. As expected, we observe that beyond some dimension, the pre-training phase becomes less effective which we attribute to its relatively naive sampling. Future work will study more sophisticated sampling schemes for training flow-based L2O frameworks. Table 6 shows the number of iterations, function evaluations and gradient evaluations for each method in the standard optimization benchmark experiments. Finally, 7 varies the training and evaluation dimensions for the dimension-agnostic experiments showing that if the dimension for training is too low, then performance will degrade as the evaluation dimension increases.

*Table 2.* **Ablation on hyperparameter $\alpha$ in training loss term $\ell_k(\mathcal{X}^k) = \alpha \text{Best}(f(\mathcal{X}^k)) + (1 - \alpha)\text{Mean}(f(\mathcal{X}^k))$.**

| Problem Class | Hyperparameter $\alpha$ | 0.0 | 0.2 | 0.4 | 0.6 | 0.8 | 0.9 | 1.0 |
|---|---|---|---|---|---|---|---|---|
| Ackley 20D | Avg. $\lvert\text{Best}(f(\mathcal{X}^K)) - f^*\rvert /\ \lvert\text{Best}(f(\mathcal{X}^0)) - f^*\rvert$ | 0.23 | 4.8e-2 | 7.3e-3 | 4.9e-3 | **3.6e-3** | 4.3e-3 | 0.054 |
| Rastrigin 20D | Avg. $\lvert\text{Best}(f(\mathcal{X}^K)) - f^*\rvert /\ \lvert\text{Best}(f(\mathcal{X}^0)) - f^*\rvert$ | 0.69 | 6.8e-3 | 5.8e-4 | 6.1e-4 | 3.0e-4 | **2.8e-4** | 0.72 |
| Levy 20D | Avg. $\lvert\text{Best}(f(\mathcal{X}^K)) - f^*\rvert /\ \lvert\text{Best}(f(\mathcal{X}^0)) - f^*\rvert$ | 0.08 | 2.7e-3 | 8.1e-4 | 5.7e-4 | **2.5e-4** | 2.7e-3 | 6.3e-2 |

*Table 3.* **Ablation on population size $P$.** The average $\lvert\text{Best}(f(\mathcal{X}^K)) - f^*\rvert /\ \lvert\text{Best}(f(\mathcal{X}^0)) - f^*\rvert$ is reported in each case.

| Problem Class | | Population size $P$ | | | |
|---|---|---|---|---|---|
| | 5 | 10 | 20 | 30 | 50 |
| Ackley 20D | 8.2e-2 | 2.3e-2 | 3.6e-3 | 2.6e-3 | 1.1e-3 |
| Rastrigin 20D | 0.21 | 8.4e-2 | 3.0e-4 | 2.4e-4 | 1.7e-4 |
| Levy 20D | 9.3e-2 | 6.1e-2 | 2.5e-4 | 1.1e-4 | 7.6e-5 |

*Table 4.* **Comparison on using the Gaussian base sampler.** The average $\lvert\text{Best}(f(\mathcal{X}^K)) - f^*\rvert /\ \lvert\text{Best}(f(\mathcal{X}^0)) - f^*\rvert$ is reported.

| Problem Class | Gaussian Base Sampler | Metric |
|---|---|---|
| Ackley 20D | Yes | 5.9e-3 |
| | No | 3.6e-3 |
| Rastrigin 20D | Yes | 1.8e-4 |
| | No | 3.0e-4 |
| Levy 20D | Yes | 6.3e-3 |
| | No | 2.5e-4 |

*Table 5.* **Performance and training efficiency comparison between pre-training and fine-tuning.**

| Problem Class | Metric | Pre-training | Fine-tuning |
|---|---|---|---|
| Ackley 2D | Number of epochs
Training time per epoch
Avg. $\lvert \text{Best}(f(\mathcal{X}^K)) - f^* \rvert \, / \, \lvert \text{Best}(f(\mathcal{X}^0)) - f^* \rvert$ | 100
0.2s
0.05 | 2000
0.8s
1.5e-3 |
| Ackley 5D | Number of epochs
Training time per epoch
Avg. $\lvert \text{Best}(f(\mathcal{X}^K)) - f^* \rvert \, / \, \lvert \text{Best}(f(\mathcal{X}^0)) - f^* \rvert$ | 100
0.2s
0.08 | 2000
0.8s
2.8e-3 |
| Ackley 10D | Number of epochs
Training time per epoch
Avg. $\lvert \text{Best}(f(\mathcal{X}^K)) - f^* \rvert \, / \, \lvert \text{Best}(f(\mathcal{X}^0)) - f^* \rvert$ | 100
0.3s
0.12 | 2000
1.0s
3.1e-3 |
| Ackley 20D | Number of epochs
Training time per epoch
Avg. $\lvert \text{Best}(f(\mathcal{X}^K)) - f^* \rvert \, / \, \lvert \text{Best}(f(\mathcal{X}^0)) - f^* \rvert$ | 100
0.4s
0.17 | 2000
1.1s
3.6e-3 |
| Ackley 50D | Number of epochs
Training time per epoch
Avg. $\lvert \text{Best}(f(\mathcal{X}^K)) - f^* \rvert \, / \, \lvert \text{Best}(f(\mathcal{X}^0)) - f^* \rvert$ | 100
0.4s
0.47 | 2000
1.3s
9.9e-3 |

*Table 6.* **Number of iterations, function evaluations and gradient evaluations for experiments in Fig. 4**

| Problem Class | Metric | RS | CEM | CMA-ES | MSGD | MSNAG | L2O-GD | **FlowOptimizer** |
|---|---|---|---|---|---|---|---|---|
| Ackley 20D | Iterations
Function evals
Gradient evals | 78
1560
- | 57
1140
- | 54
1080
- | 73
1460
1460 | 72
1440
1440 | 73
1460
1460 | 10
200
200 |
| Rastrigin 20D | Iterations
Function evals
Gradient evals | 165
3300
- | 103
2060
- | 93
1860
- | 163
3260
3260 | 160
3200
3200 | 163
3260
3260 | 10
200
200 |
| Levy 20D | Iterations
Function evals
Gradient evals | 57
1140
- | 47
940
- | 44
880
- | 51
1020
1020 | 50
1000
1000 | 51
1020
1020 | 10
200
200 |

*Table 7.* **Varying training and validation dimensions on Ackley.** The average $\lvert \text{Best}(f(\mathcal{X}^K)) - f^* \rvert \, / \, \lvert \text{Best}(f(\mathcal{X}^0)) - f^* \rvert$ is reported.

| Training dim. | Validation dim. | | | |
|---|---|---|---|---|
| | 5 | 20 | 50 | 200 |
| 5 | 1.3e-2 | 0.76 | 1.89 | 2.74 |
| 20 | 7.9e-3 | 1.7e-2 | 7.4e-2 | 0.15 |

# D. Additional Discussion

## D.1. Discussion on Potential Performance Guarantees

Providing formal performance guarantees for L2O methods is inherently challenging, as the motivation behind L2O is to replace fixed algorithmic choices with learnable components (neural networks), in order to improve empirical performance (Amos et al., 2023). This added flexibility, however, often comes at the expense of classical algorithmic guarantees. Nevertheless, several research directions have emerged to formalize the behavior of L2O methods, including safeguarding mechanisms (Heaton et al., 2023), worst-case guarantees (Banert et al., 2024; Sambharya et al., 2025; Martin et al., 2025) and probabilistic generalization bounds (Saravanos et al., 2025; Sambharya & Stellato, 2025; Sucker et al., 2025).

In the following, we briefly discuss how generalization guarantees could be established for FlowOptimizer. In particular, the same general framework used to derive generalization bounds in (Saravanos et al., 2025) can be applied here, provided that

a suitable performance metric is selected for the non-convex setting. A natural choice would be

$$q = \min\left\{ \frac{f(x_{\text{best}}^K(\zeta;\theta)) - f^*(\zeta)}{f(x^0(\zeta)) - f^*(\zeta)},\ 1 \right\} \tag{38}$$

which quantifies the final progress relative to initialization, where $\zeta$ and $\theta$ denote the problem instance and neural network parameters, respectively. Since this metric satisfies the criteria discussed in Section 5 of (Saravanos et al., 2025), the same argument can be used to obtain a generalization bound analogous to Theorem 3 in that work, after replacing their progress metric with (38). Following the same training and evaluation procedure, would allow us to establish generalization bounds for FlowOptimizer. We note that such a bound would rely on either knowing the optimal objective value (not the optimal solution) or at least having a sufficiently accurate estimate of it. We leave the design and evaluation of more practical bounds that are suitable for non-convex optimization for future work.

