# OpenReview forum: "Learning-to-Optimize via Deep Unfolded Flows"
_ICML.cc/2026/Conference — ICML 2026 spotlight_

### Official Review · Reviewer_g3Hb · 2026-02-25

**Soundness:** 3
**Presentation:** 4
**Significance:** 3
**Originality:** 3
**Overall Recommendation:** 4
**Confidence:** 4

**Summary:**

This paper proposes a deep unfolded, flow-based framework to accelerate the solution of unconstrained optimization problems. The approach models optimization iterations using a flow model, integrating learn-to-optimize strategies with sampling techniques. Additionally, the authors conducted a two-stage training procedure for the underlying self-attention network, consisting of pre-training followed by end-to-end fine-tuning. Numerical results are provided to support the claim.

**Compliance With Llm Reviewing Policy:**

Affirmed.

**Key Questions For Authors:**

My main concerns are on the empirical feasibility of the proposed framework.
1. In Algorithm 2, the total loss is accumulated across all unrolled layers. While this is reasonable, it may demand an extremely large amount of computational resources when scaling to larger problems or requiring a larger $d$. Moreover, the value of $K$ is not clearly specified or discussed. A brief explanation of how $K$ and related hyperparameters were chosen would be helpful.
2. The experiments are limited to problems of small scale (10–200). Since I am not deeply familiar with standard unconstrained optimization benchmarks, it would be helpful if the authors could justify why these particular problem sizes were selected.
3. Figures 4 and 5 suggest promising generalization to larger instances. However, because performance is reported in absolute error, it is difficult to judge whether the behavior on 200D instances is truly comparable to that on smaller ones. Reporting relative error would make cross-dimensional comparisons much clearer.
4. Following Q3, if the performance remains consistent across 10D and 200D, could it indicate that the chosen dimension $d$ is a "overkill" for smaller problems? A short discussion about $d$ would be valuable.

**Limitations:**

yes

**Strengths And Weaknesses:**

Strength:
1. The idea of using a flow model with sampling-based optimization is promising and offers a novel perspective on unconstrained optimization.
2. The presentation is clear, well-structured, and easy to follow.

Weakness:
1. The proposed training setting appears to lack scalability.
2. The experiments are conducted on rather toy-like examples.
3. Some minor typos, such as in page 4 "] We then apply self-"

---

> ### Author Rebuttal · Authors · 2026-03-31
>
> We are thankful to the reviewer for their thoughtful and constructive feedback, which we have carefully addressed below.
>
> # Response to Weaknesses
>
> 1. We thank the reviewer for raising this point and address it in detail in Question 1 below. In brief, the computational cost of deep unfolding is a training-time concern only and does not affect significantly inference cost or optimization performance.
>
> 3. We appreciate this feedback and **will better emphasize the difficulty and relevance of the benchmarks in the final version of our paper**.
>
>     The **classical optimization benchmarks** (Ackley, Rastrigin, and Lévy) are **very highly non-convex test functions** widely used in the optimization community [R1-R3]. They are explicitly designed to challenge both gradient-based and sampling-based methods, with exponentially many local minima, **making them notoriously difficult especially under limited iterations**.
>
>
>     Beyond these benchmarks, we include **three real-world problems**: (i) robotic arm inverse kinematics with obstacle avoidance; (ii) power grid economic load dispatch, a classical industrially relevant benchmark featuring valve-point effects and severe multimodality; and (iii) supply chain facility location (Multi-Source Weber Problem), arising in logistics and operations research.
>
>     [R1] Hansen and Ostermeier. "Completely derandomized self-adaptation in evolution strategies." Evolutionary computation (2001).
>
>     [R2] Kochenderfera and Wheeler. Algorithms for optimization. MIT Press, 2019.
>
>     [R3] Chen et al. "Learning to optimize: A primer and a benchmark." JMLR (2022).
>
> 3. We thank the reviewer for their thoroughness and will make sure any typos are fixed.
>
> # Response to Questions
>
> 1. We use $K = 10$ unfolded optimization iterations accross all experiments, a reasonable number typically needed for achieving a satisfying accuracy with FlowOptimizer. Each iteration includes a lightweight Euler ODE integration with $T = 5$ integration steps. This setup mantains substantial computational tractability as fine-tuning takes only approx. 10-20 mins on a single RTX 4090 for all problems.
>
>     As mentioned in the limitations paragraph, we acknowledge that extending to larger K or larger problem dimensions $d$ will increase the length of the gradient graph through the ODE solver, which is the primary computational bottleneck during training. At inference, FlowOptimizer simply executes $K$ sequential flow model evaluations, which is lightweight. Yet as discussed in our response to reviewer 46ys, the same model can be applied for larger $K$ during inference.
>
> 2. The chosen scales are standard and meaningful for the benchmark and application domains considered. For non-convex benchmarks like Ackley, Rastrigin and Lévy, $d=20$ is a widely used evaluation setting that remains genuinely hard for sampling-based methods due to exponential volume growth and multimodality. For the real-world problems, the dimensions are dictated by the physical structure of each application. Finally, the dimension-agnostic variant is evaluated across $d=20$ to $d=200$, well beyond the training regime, demonstrating strong generalization to **substantially higher-dimensional instances where sampling-based methods struggle**.
>
> 3. We note that for all standard benchmarks considered (Ackley, Rastrigin, Lévy), the ground-truth optimal value is known. Therefore, the absolute error $|\text{Best}(f(X^K)) - f^{\star}|$ already directly measures progress in a meaningful and interpretable way. We are also happy to additionally report relative error normalized by the initial population error $|f(X^0) - f^*|$ (as we do in Table 1 for the real-world problems) in the final version if the reviewer deems it helpful.
>
> 4. We thank the reviewer for this insightful question. The consistent performance
> across dimensions is not an indication that smaller training dimensions are
> "overkill"; rather, it reflects the effectiveness of the **dimension-agnostic
> architecture** (Sec. 3.2), which **learns geometric optimization patterns**
> (e.g., how to coordinate a population to escape local basins) that **naturally
> transfer across dimensions**. Importantly, the training dimension does matter:
> as shown in Table 7, a model trained on $d=20$ generalizes substantially
> better than one trained on $d=5$ across all validation dimensions, including
> $d=200$. This suggests that **a sufficiently rich training dimension is necessary
> to capture the complexity of the optimization dynamics**, and that **$d=20$ strikes
> a good balance between training cost and generalization quality**.
>
>     *Added experiment:* Table 7 in https://figshare.com/s/f54eeab07c6a66a5dfd3
>
> We appreciate the reviewer’s feedback and believe these clarifications strengthen the paper. In light of these clarifications and new experiments, we hope the reviewer will consider increasing their evaluation. We remain available for any further questions or discussion.

---

> > ### Author Rebuttal · Reviewer_g3Hb · 2026-04-02
> >
> > My questions and concerns are properly addressed, and I choose to remain my score.

---

> > > ### Author Response · Authors · 2026-04-02
> > >
> > > We sincerely appreciate the reviewer's positive assessment of our paper and we are glad to see that their comments have been fully addressed.

---

### Official Review · Reviewer_5HHj · 2026-03-11

**Soundness:** 3
**Presentation:** 3
**Significance:** 3
**Originality:** 3
**Overall Recommendation:** 5
**Confidence:** 4

**Summary:**

This paper proposes **FlowOptimizer**, a flow-based learning-to-optimize (L2O) pipeline that learns to adapt ensembles in order to provide a velocity-field-based iterative update for general non-convex optimization problems. The method consists of two phases of offline training. In the pretraining stage, it aims to learn a particle velocity field using a flow-matching loss through a pushforward operator that encourages the movement of ensembles toward lower-potential regions. In the fine-tuning stage, a few time-integration steps of the learned velocity field are computed and trained using a conventional L2O target loss, defined as a convex combination of mean and best performance. The authors report results not only on conventional nonlinear optimization tasks (including Ackley, Rastrigin, and Levy), but also on several real-world application problems (ranging from 10 to 80 dimensions) to demonstrate the effectiveness of the proposed algorithm.

**Compliance With Llm Reviewing Policy:**

Affirmed.

**Final Justification:**

Authors addressed my concerns and added additional reproducibility experiments. Although they have not replied to my further response by the moment I offer the final justification, since all further pinpoints are suggestions to make paper better not technical drawbacks, I would raise my scores from weak accept to accept. If authors can complete a round of revision or reply to my suggestions in time it would be cool though. >>>>>>> Authors did respond in last minutes, and this is a solid answer. Please incorporate them in final camera-ready version.

**Key Questions For Authors:**

* What is the design principle behind the unrolling weights $w_k$? I could not find any specification regarding the choice of $w_k$, which is important for reproducibility. Are these weights tuned case by case, or are they determined by a fixed schedule across all optimization tasks?
* Following the previous question, the appendix reports a population size of $P=20$ and a balancing parameter of $\alpha=0.8$. Were these hyperparameters tuned, or were they chosen as reasonable defaults? Would it be possible to include ablation studies to clarify their role in the method? For example, would using more particles provide a more accurate estimate of the mean and covariance context?
* I suggest that the authors provide trajectory visualizations for 2D illustrative examples to support the numerical claim that FlowOptimizer achieves very fast convergence. In particular, it would be helpful to visualize the learned velocity field directly.
* For each experiment, could the authors additionally report results using the best-performing optimizer for that problem with $P$ random initial points, so that readers can better assess whether the proposed L2O method is competitive with optimizers that are specialized for task-specific solutions?
* In Lines 553–555, the authors state that “a Gaussian base sampler was used for the Rastrigin problems, as it was observed to facilitate exploration.” Why is a biased sampler choice introduced here, whereas all other methods appear to follow the alternative rule of using the “previous population”?

**Limitations:**

Yes.

**Strengths And Weaknesses:**

Developing an impactful L2O method that can adapt to general non-convex optimization problems is known to be difficult. Scalability and feasibility both remain major challenges, as L2O methods typically require an intensive amount of training to learn how to optimize a specific class of task functionals, whereas a fixed random initialization policy combined with Adam/Muon in neural network training often already performs well. This paper attempts to provide a reasonable solution to these challenges. Although the proposed method is still evaluated primarily on relatively low-dimensional optimization tasks, implementing such an approach is already nontrivial, and I appreciate the authors’ effort in making the idea of learning a velocity field tangible. The effectiveness demonstrated in several real-world examples also strongly supports its practical applicability. In particular, the reported generalization from 20 dimensions to 200 dimensions using a dimension-agnostic shared context encoder for velocity field prediction is a notable takeaway and suggests promise for higher-dimensional generalization.

Nevertheless, the lack of a reproducibility statement — including clearer experimental setup details, justification of design choices, and a commitment to release code upon acceptance — currently harms the overall quality of the paper. These concerns are elaborated in the next section, and I hope the authors can address them. Additionally, the paper does not provide any theoretical understanding of whether the method can guarantee asymptotic convergence (cf. [1]) or offer any form of generalization bound (cf. [2]). I would also note that this paper appears closely related to the literature on particle swarm optimization (PSO), yet discussion of the connection to PSO is currently missing.

[1] Liu, Jialin, et al. “Towards constituting mathematical structures for learning to optimize.” *International Conference on Machine Learning*. PMLR, 2023.

[2] Shi, Mingjia, et al. “Make optimization once and for all with fine-grained guidance.” *arXiv preprint* arXiv:2503.11462 (2025).

---

> ### Author Rebuttal · Authors · 2026-03-31
>
> We thank the reviewer for recognizing the novelty of our work and for the thoughtful feedback, which we address below.
>
> # Response to Weaknesses
>
> 1. **Reproducibility.** We commit that **any missing experimental details will be fully documented in the appendix and code will be released with the final version of the paper**.
>
> 2. **Theoretical guarantees via generalization bounds.** We thank the reviewer for providing us with the opportunity to discuss about potential guarantees for the FlowOptimizer. First, it is worth noting that L2O methods generally trade the asymptotic convergence guarantees of classical handcrafted optimizers for substantially improved empirical performance; a well-recognized tradeoff in the L2O literature. Nevertheless, **our method is directly compatible with several generalization bounds established for L2O methods**, such as in [R1], [R2, Section 5], [R3, Section 4.3]. We would be happy to include a brief illustration if the reviewer deems it appropriate.
>
>
>     [R1] Sambharya & Stellato, "Data-Driven Performance Guarantees for Classical and Learned Optimizers.", JMLR (2025).
>
>     [R2] Saravanos et al.,  "Deep Distributed Optimization for Large-Scale Quadratic Programming.", ICLR 2025.
>
>     [R3] Oshin et al., "Deep FlexQP: Accelerated Nonlinear Programming via Deep Unfolding.", ICLR 2026.
>
>
> 3. **Connection to PSO.** We thank the reviewer for bringing up this important connection. While PSO methods and the FlowOptimizer both maintain a population of candidates, the key distinction is that **PSO relies on fixed, hand-designed velocity update rules**, whereas **FlowOptimizer learns flow velocity fields that capture population-level dynamics from data**. **FlowOptimizer can thus be seen as a richer and more expressive framework that subsumes the population-based intuition of PSO**. We will add a discussion of this connection in the related work.
>
> # Response to Questions
>
> 1. The **unrolling weights** follow a **linear ramp** normalized to sum to 1, i.e., $w_k = \frac{2k}{K(K+1)}$, assigning greater importance to later iterations. We will add this specification to the paper to ensure full reproducibility. These weights are **fixed across all problem classes**. We tried alternative schedules (quadratic, exponential, uniform) and found the method robust to this choice.
>
> 2. The **balancing parameter** $\alpha = 0.8$ and **population size** $P = 20$ were chosen as reasonable defaults after light tuning and kept **fixed across all problem classes**.
>     - $\alpha$: values near $0$ risk mode collapse, while values near $1$ reduce diversity. We found that $\alpha = 0.8$ strikes a balance that empirically works robustly across all benchmarks.
>     - $P$: larger populations provide richer estimates of the mean and covariance context at increased computational cost, and $P = 20$ was found to offer a  good trade-off. We provide the following **ablation studies varying $\alpha$ and $P$  on the Ackley, Rastrigin and Lévy 20D problems**, reporting both best and mean objective  values.
>
>     *Added experiment:* Tables 1 and 2 in https://figshare.com/s/f54eeab07c6a66a5dfd3
>
>
> 3. We thank the reviewer for this suggestion. We have added **2D trajectory visualizations on the Ackley function** for all methods, as well as a direct visualization of the **learned velocity field** of FlowOptimizer. These figures clearly illustrate the fast convergence of FlowOptimizer's population dynamics compared to all baselines, and provide intuition for how the learned velocity field adapts based on the current population state and contextual information.
>
>     *Added 2D visualizations:* https://figshare.com/s/355417c2ca4ba33d6578
>
> 4. If we correctly understand the question, the best-performing classical baseline for each problem can already be read directly from Figs. 4, 5 and Table 1, using random initial points. We are happy to make this comparison clearer in the revision, and would welcome any clarification if the reviewer had a different comparison in mind.
>
> 5. The base sampler is an **optional architectural component** (Section 3.2). For Rastrigin, its periodic structure risks population collapse, and the Gaussian sampler introduces controlled diversity, while the update rule remains fully learned. An ablation confirms: (i) **FlowOptimizer without base sampler still outperforms all baselines on Rastrigin**, and (ii) **no base sampler yields better performance on all other problems, consistent with the learned transport being effective on directly transporting the populations**.
>
>     *Added experiment:* Table 3 in previous link.
>
>
> We sincerely thank the reviewer for their constructive feedback and suggestions. We have carefully addressed all points and will incorporate these improvements in the final version.
>
> In light of these clarifications and revisions, we hope the reviewer will consider increasing their evaluation. We remain available for any further questions or discussion.

---

> > ### Author Rebuttal · Reviewer_5HHj · 2026-04-01
> >
> > I appreciate author's response. I am inclined to further raise my score to accept. However, I have some further follow up clarification point and suggestions to authors.
> >
> > > Nevertheless, our method is directly compatible with several generalization bounds established for L2O methods, such as in [R1], [R2, Section 5], [R3, Section 4.3]. We would be happy to include a brief illustration if the reviewer deems it appropriate.
> >
> > Only R1 may be a straight adaptation as it discusses the general case. I want to understand in author's mind, how to adapt the conclusion from deep unfolded QP solvers in R2 and R3 to the proposed FlowOptimizer?
> >
> > > We have added 2D trajectory visualizations on the Ackley function for all methods, as well as a direct visualization of the learned velocity field of FlowOptimizer.
> >
> > The figure is hard to read and the domain scale is somehow $[-30,30]^2$, whereas in manuscript the Ackley function's domain is $[-4,4]^2$. Please plot in $[-4,4]^2$ only. I would also suggest a trajectory or streamline plot for some particles rather than snap shot of ensemble of particles to further improve the visualization. If one can also plot other method's trajectory that would be a good comparison and illustration as well.
> >
> > >  If we correctly understand the question, the best-performing classical baseline for each problem can already be read directly from Figs. 4, 5 and Table 1, using random initial points. We are happy to make this comparison clearer in the revision, and would welcome any clarification if the reviewer had a different comparison in mind.
> >
> > For three realistic examples, are all compared method being hyper-tuned to show its optimal performance? Further, have authors tried MS-Adam, MS-SQP, or some task specific optimizer literature have proposed?

---

> > > ### Author Response · Authors · 2026-04-08
> > >
> > > We really thank the reviewer for their positive feedback, and most importantly, we are grateful for their thoughtful questions and suggestions, which have helped us further strengthen the paper.
> > >
> > > > Only R1 may be a straight adaptation as it discusses the general case. I want to understand in author's mind, how to adapt the conclusion from deep unfolded QP solvers in R2 and R3 to the proposed FlowOptimizer?
> > >
> > > We thank the reviewer for this question. Note that the approach presented in Section 5 of [R2] is in fact general to learning-to-optimize and not directly tied to QPs. The approach is valid as long as the progress metric $q$ in Eq. (8) is bounded between 0 and 1.
> > >
> > > That means that we can use the same approach for obtaining a generalization bound, but we need to select a meaningful metric for our non-convex setting. A proper choice would be
> > > $$q = \text{min} [
> > > (f(x^K_{best} (\zeta; \theta)) - f^* (\zeta) ) /
> > >   ( f(x^0 (\zeta) )- f^* (\zeta) )
> > > , 1]$$
> > > which measures the final progress relative to initialization, where $\zeta$ and $\theta$ denote the problem instance and neural network parameters, respectively. Since this metric satisfies the criteria discussed in [R2, Section 5], the same argument can be used to obtain a generalization bound analogous to Theorem 3, after replacing their progress metric with ours. Following the same training and evaluation procedure, we can compute generalization bounds for the FlowOptimizer. We note that such a bound would rely on either knowing what is the optimal objective value (not the solution) or at least having a good estimate of it. We leave further improvements on such bounds for future work.
> > >
> > > > The figure is hard to read and the domain scale is somehow $[-30,30]^2$, whereas in manuscript the Ackley function's domain is $[-4,4]^2$. Please plot in $[-4,4]^2$ only. I would also suggest a trajectory or streamline plot for some particles rather than snap shot of ensemble of particles to further improve the visualization. If one can also plot other method's trajectory that would be a good comparison and illustration as well.
> > >
> > > We thank the reviewer for these helpful suggestions regarding the visualization. First, we would like to clarify that the domain of the Ackley function is indeed $[-30,30]^2$ as stated in line 692. Figure 6 in the Appendix was zoomed into $[-4,4]^2$, only to better highlight the highly non-convex structure of the landscape.
> > >
> > > Per the reviewer's request, we have added several new figures to improve the presentation of these 2D examples.
> > >
> > > *Added figures:* Figures 1 and Figure 2 in https://anonymous.4open.science/r/icmlrebuttal2026-DD41/ show the velocity field more clearly for full domain $[-30,30]^2$ and zoomed-in region $[-4,4]^2$, respectively. Figure 3 shows the full trajectories of three particles for each method. Finally, Figures 4 and 5 show population snapshots for all methods over the full and zoomed-in domains, respectively.
> > >
> > >
> > >
> > > > For three realistic examples, are all compared method being hyper-tuned to show its optimal performance? Further, have authors tried MS-Adam, MS-SQP, or some task specific optimizer literature have proposed
> > >
> > > We thank the reviewer for these valuable suggestions and clarification requests.
> > >
> > > - Yes, for the three real-world examples, **all baselines methods are hyperparameter-tuned**. The scanned hyperparameters for each method are provided in the following table.
> > > *Added table:* Table 1 in previous link.
> > > - **We have added MS-Adam and MS-Newton** as additional baselines in our experiments across all problems. Since our setting is unconstrained, the suggested SQP baseline reduces to Newton's method. Across all experiments, FlowOptimizer continues to achieve the best overall performance. Note that there also exist other specialized constrained optimization methods for addressing the real-world non-convex problems before relaxing constraints (e.g., sequential convex programming, interior point, etc.) but such methods would far exceed the computational budget in our experiments.
> > > *Added tables:* Tables 2 and 3 in previous link.
> > >
> > > Overall, we are sincerely grateful to the reviewer for their positive evaluation, as well as for their detailed and constructive suggestions. We will incorporate all of these improvements in the final version of the paper.

---

### Official Review · Reviewer_F8V8 · 2026-03-12

**Soundness:** 3
**Presentation:** 3
**Significance:** 3
**Originality:** 4
**Overall Recommendation:** 5
**Confidence:** 3

**Summary:**

This paper introduces a novel approach in the context of Learning to Optimize, where Flow models are trained on general optimization problems to improve a population of candidate solutions over time, by conditioning the model on contextual information (objective values, gradient information, population statistics).
The proposed method is able to scale on higher dimensions and it is compared with classical gradient-based and sample-based approaches.

**Compliance With Llm Reviewing Policy:**

Affirmed.

**Final Justification:**

The authors addressed my main concerns, so I think the contribution and the method are strong enough to be accepted

**Key Questions For Authors:**

1. Is it possible to adapt the method to operate (with reasonable time complexity) on problems where the objective requires a costly evaluation?

**Limitations:**

yes

**Strengths And Weaknesses:**

# Strengths:
1. The paper is well written and the idea is well explained
2. The idea is sound, original and the contribution is relevant for many settings requiring to optimize problems in low and high dimensions
3. The method tackles the optimization problem from a completely new perspective which also allows for future improvements along the same direction

# Weaknesses:
1. It looks like the method, though scalable to higher dimensions, would still be subject to some scalability issues related to sample-based methods. For example in problems where evaluating the objective function would be extremely costly (which is a problem whenever multiple individual evaluations are required for each iteration) or for extremely high dimensional spaces (for instance, in training neural networks). This may restrict the applicability of the method.

---

> ### Author Rebuttal · Authors · 2026-03-31
>
> We really appreciate the positive evaluation and are grateful the reviewer highlighted the novelty and broader relevance of our work. We also thank the reviewer for their criticism, which we have carefully addressed below.
>
> # Response to Weakness
>
> We appreciate the reviewer raising this point. We would like to highlight that FlowOptimizer directly addresses several of the scalability concerns associated with classical sampling-based methods. In particular, its amortized nature, population-size agnosticism, and dimension-agnostic architecture collectively mitigate the key limitations the reviewer identifies. We elaborate on each of these properties in our response below.
>
> # Response to Question
>
> There are various ways we can adapt the proposed method in the case we have problems where the objective requires a highly costly evaluation and to enchance scalability.
>
> a. **Population size adaptation.** A key property that perhaps was not sufficiently emphasized in the paper, is that FlowOptimizer is **agnostic to the population size** during training and evaluation. The same trained architecture can be deployed with a smaller population to reduce the number of objective evaluations per iteration during deployment, if needed. This applies for both the standard version and the dimension-agnostic variant.
>
> b. **Scalability to high dimensions.** The proposed dimension-agnostic variant enables training on low-dimensional problems and deployment to much higher-dimensional ones (Fig. 5), thereby **avoiding the need to learn directly in very high-dimensional spaces**. This is a significant advantage over sampling-based methods, which are well-known to suffer from the curse of dimensionality as dimension grows.
>
> c. **Amortized optimization.** The main computational burden is shifted to training. Once trained, the method requires only a small number of iterations to reach high-quality solutions (as seen in Figs. 4, 5 and Table 1), resulting in **fewer total objective evaluations during deployment** compared to standard sampling-based approaches.
>
> d. **Compatibility with surrogate models.** The framework can naturally be combined with surrogate or multi-fidelity models in settings where true objective evaluations are extremely costly.
>
>
> Overall, we highly appreciate the reviewer's praising comments and thoughtful criticism. We hope our response further strengthens the reviewer's positive assessment of our work.

---

> > ### Author Rebuttal · Reviewer_F8V8 · 2026-04-02
> >
> > I thank the authors for the clarifications, which I find helpful for a better understanding and I would suggest to emphasize some of these points in the main text (especially the agnostic population size).
> > I confirm my positive assessment.

---

> > > ### Author Response · Authors · 2026-04-02
> > >
> > > We are thankful to the reviewer for their positive assessment and valuable feedback. We will incorporate the suggested clarifications in the final version.

---

### Official Review · Reviewer_46ys · 2026-03-13

**Soundness:** 3
**Presentation:** 3
**Significance:** 3
**Originality:** 3
**Overall Recommendation:** 4
**Confidence:** 4

**Summary:**

This paper introduces a new method for learning to optimize (L2O). The authors first discuss the complementary aspects of traditional sampling-based optimizer and existing L2O optimizer, and then set the research motivation of this paper as finding a interpolation between these two paradigms. Specifically, the authors propose a auto-regressive architecture which is based on self-attention, in this architecture the decision variables, fitness values, gradient information of candidate population could be encoded in a generalized way. Then the authors propose a two-stage training framework to train such learnable optimizer. In the first stage, flow matching is used to learn velocity field between current population and its potential improvement direction in a self-supervised fashihon. After this pre-training, the learned basic neural policy undergoes the second fine-tuning stage, where it is unfolded along the optimization horizon, and trained to minimize the objective value of the training opitmization problems at each optimization steps by gradient descent. The experiments include both synthetic validation and realworld scenario validation, which show that the proposed method is superior to several representative L2O baselines, as well as robust CMAES optimizer.

**Compliance With Llm Reviewing Policy:**

Affirmed.

**Final Justification:**

The authors have proposed an interesting work, and all of my previous concerns have been addressed.

**Key Questions For Authors:**

1. lines 207-208, right column. I am confused here, what do the authors mean in the equation where two vectors ($v_{j,i}^{loc}$ and $<a,b>$) with different dimension are added, how they are added together?

2. In Sec. 3.3, for a candidate population, a much larger population with $\gamma P$ size is sampled for learning the flow matching, the $\gamma$ is set to what value? one more question, as the authors claimed, $\gamma >> 1$, does this indicate that the proposed mathod is sample-inefficient at all?

3. In Sec. 4.1, I think the proposed method might be cursed by the optimization horizon T. Since the authors stated that for each optimization generation step $t \in [0,T]$, they initialize a pre-trained flow-matching network, then we can imagine that we must fix the optimization horizon. How is this possible for diverse optimization problems, for those simpler ones, one may solve them by 50-100 optimization steps, in contrast, for more complex ones, we may need thousands of steps.

4. Considering the fine-tuning stage (Sec. 4.2), for those black-box optimization problems, or those whose objective function is not differentiable, how you train (or finetune) your model using Eq. (13)? That is, the proposed method has very strong dependence on the target optimization problems.

5. The experimental protocol is not clear at all. For all validation test, what is your training set? What are the values selected for all hyper-parameters? Are they at least selected by grid search?

6. How many function evluations your method require to achieve the performance in Fig. 4? You only report the performance curve under the wall-clock timeline, however, in practice, many problems are expensive, so report number of function evaluations is important.

7. I have an additional interesting questions, I will do my best to describe this question (while it may be hard to explain). Consider two different optimziation problem instances, they might share part of optimization landscape, while the optimal and other landscape parts are totally different. Now consider the proposed method in this paper, it sorely relies on the sample points to determin the state, which is used to dictate next generation sample points. If at some time step, the two populations on these two different problems are identical, then the dictated new population must be identical (youe model is deterministic), then how we ensure this fixed model could solve both of these two problems (since their optimal might be in totally opposite direction). This is just an in-depth question I want to discuss with the authors, not a critical point.


To summarize, while this paper is interesting with some novel aspects, the missing details hinder me from accepting it as a top-tier conference paper. I hope the authors could carefully address my concerns and questions. My score will be adjusted accordingly.


=====================
I increase my score.

**Limitations:**

The authors have provided a discussion on the limitations in the end of this paper.

**Strengths And Weaknesses:**

Strengths:

1. I think this paper is a smooth and continuous effort for addressing the low-effectiveness issue in existing L2O works. The motivation is clear, and the methodology (self-supervised flow matching + post fine-tuning) is an novel and useful (from my angle) update to existing works.

2. Overall, the writing of this paper is clear, with each key concept well explained. The methodology is genrally well organized and elaborated.

3. From the experimental results provided by the authors at hand, I think the proposed two-stage training is interesting and solid.

However,  I have following concerns and questions, which I hope the authors could discuss with me in the rebuttal phase.

Weaknesses:

1. First of all, when we disicuss about learning to optimize, there is another research track termed as "Meta-Black-Box Optimization" (https://ieeexplore.ieee.org/abstract/document/10993463) that should be considered as direct related works or comparison baselines. This field also explores how to learn effective, generalizable and efficient optimziation policies through meta-learning. I list several works in this field that show very similar idea with this paper: i) https://ojs.aaai.org/index.php/AAAI/article/view/34036; ii) https://openreview.net/forum?id=vLLYlSK6qJ; iii) https://openreview.net/forum?id=vLJcd43U7a. I hope the authors could position this paper more objectively considering these Meta-Black-Box Optimization works.

2. I found the core idea in your feature embedding (page 4) is actually not new to L2O or Meta-Black-Box Optimization field. In some existing works, such as: i) https://dl.acm.org/doi/abs/10.1145/3638529.3653996; ii) https://dl.acm.org/doi/abs/10.1145/3712256.3726309; iii) https://openreview.net/forum?id=EEI5R89Cmv; iv) https://ieeexplore.ieee.org/abstract/document/11297580. Consider providing a thorough discussion on the similarity and differences between them and your proposed one.

3. In its current status, this paper lacks many details in both the methodology and implementation. I leave these detailed questions in the Key Question section.

---

> ### Author Rebuttal · Authors · 2026-03-31
>
> We thank the reviewer for the valuable feedback and address all points below.
>
> # Response to Weaknesses
>
> 1. We thank the reviewer for bringing up "Meta-Black-Box Optimization" (MBBO). **MBBO typically studies non-differentiable objective functions with costly evaluations (via black-box simulators, etc.)**, using particle-based methods without derivatives **under a function evaluation budget**. In contrast, L2O typically assumes **differentiable objectives** and aims for low **actual computational (wall-clock) time**. Our work belongs in the second class, but you are right that there is a connection to the first class as well, since FlowOptimizer learns velocity fields that operate on populations. **The final version of our paper will include a discussion on similarities with MBBO.**
>
> 2. We do not claim feature embedding or permutation-invariant attention to be a core novelty of this paper; no such claim is made in the abstract or contributions. The **fundamental novelty of our work** is to **learn an iterative deep unfolded optimizer via a flow velocity field representation** towards **combining the best of both gradient and sampling-based optimization**. Within this setup, we have found that an attention-based mechanism can be very effective in preserving permutation invariance and extracting useful information. **We will discuss prior works that have used similar neural network architectures in the final version.**
>
> 3. See Question 5.
>
> # Response to Questions
>
> 1. The shared per-dimension decoder in Eq. (8) outputs a **scalar** value $v_{j,i}^{loc}$, which is added in Eq. (9) to the inner product $\langle a_{j,i},\, b_j\rangle$ of two vectors, which is also a scalar.
>
> 2. The **pre-training phase via flow matching** is **"simulation-free"**, making it more computationally and memory-efficient. We use $\gamma = 20$ in all our experiments. Nevertheless, as correctly pointed out by the reviewer, if the dimension increases, we would need to sample more during training (offline), making pre-training most beneficial at moderate dimensions. Beyond a certain dimension, the fine-tuning stage alone is preferable. We provide a comparison across dimensions.
>
>     *Added experiment:* Table 4 in https://figshare.com/s/f54eeab07c6a66a5dfd3
>
> 3. Note that the flow integration time $t \in [0,1]$ is internal to each iteration $k$ and governs the ODE integration; it is not the optimization horizon. The optimization horizon is $K$, the number of unfolded iterations. We use $K=10$ across all experiments.
>
>     That said, you are indeed raising a valid point of how can we adapt the method if we need more iterations during deployment. A well-established adaptation in the deep unfolding literature is to share parameters across the final $K_{\text{final}}$ iterations, which is precisely what we do (Sec. 4.1). In all our experiments, we set $K_{\text{final}} = 3$. **This allows the model to be executed for an arbitrary number of additional iterations** at inference time beyond $K$ without introducing new parameters. We would be happy to include an experiment demonstrating this if the reviewer deems it appropriate.
>
> 4. We assume the objective functions are differentiable; this will be clearly stated in the final version. Dependence on training problems is common in L2O, but **we show that policies trained on one problem class remain effective on others.**
>
>     *Added experiment:* Table 5 in previous link.
>
> 5. **We commit to fully documenting all experimental details and releasing code with the final version.** See also Q1, Q2 responses to Reviewer 5HHj and added ablation tables. For training and validation sets we used 1000 and 300 problems from each class in every experiment. All baseline methods were tuned through grid search.
>
> 6. Function and gradient evaluation counts for all methods in Fig. 4 are reported below.
>
>     *Added table:* Table 6 in previous link.
>
>
> 7. We thank the reviewer for this insightful example. Identical populations can only produce identical outputs on the first iteration; in subsequent iterations, historical context differentiates the trajectories. We have constructed the following experiment to study this case: We constructed an experiment with 5 problem instances **sharing identical initial populations in a flat region**: as expected, **iteration 1 produces identical populations**, but the model has learned to spread particles in flat regions, and **by iteration 4 the populations are substantially different**, eventually **converging to the correct optimum**. This is certainly a very valuable exposition that will be included in the final version.
>
>     *Added experiment:* https://figshare.com/s/83fc7d8872c1deb12629
>
> We have carefully addressed all raised points and believe incorporating them will further strengthen the paper. In light of these clarifications and added results, we hope the reviewer will consider increasing their evaluation. We remain available for any further discussion.

---

> > ### Author Rebuttal · Reviewer_46ys · 2026-04-01
> >
> > I appreciate the authors' responses. Should they promise to make all suggested revisions into the final paper, I will increase my score accordingly.

---

> > > ### Author Response · Authors · 2026-04-02
> > >
> > > We really thank the reviewer for all their valuable feedback and insights which will further enhance the quality of our paper.
> > >
> > > We are glad that your comments have been fully addressed and we commit that all suggested revisions will be incorporated into the final paper.

---

### Decision · Program_Chairs · 2026-04-30

**Decision:**

Accept (spotlight)

**Comment:**

This paper proposes **FlowOptimizer**, a learned iterative optimization framework based on deep unfolded flow dynamics over a population of candidate solutions. The method combines a self-supervised pretraining stage, where a velocity field is learned via flow matching, with an end-to-end fine-tuning stage that directly optimizes objective values over a class of target problems.

The reviewers were consistently positive about the paper’s core contribution. They viewed the idea of representing optimization as a learned flow over populations as novel and promising, and found the two-stage training framework technically sound and well motivated. The empirical results, spanning both standard non-convex benchmarks and several real-world problems, were seen as strong evidence that the approach is practically effective.

The main concerns raised during review were about positioning, reproducibility and scalability, they are properly addressed in the rebuttal phase.

This is a strong paper with a clear conceptual contribution, convincing empirical evidence, and broad potential impact at the intersection of optimization and generative modeling. The final version should incorporate the promised clarifications and additional experimental details.